# Efficient Knowledge Injection in LLMs via Self-Distillation

**Kalle Kujanpää**                                    *kalle.kujanpaa@aalto.fi*
**Pekka Marttinen**                                   *pekka.marttinen@aalto.fi*
*Department of Computer Science, Aalto University*
*Finnish Center for Artificial Intelligence (FCAI)*

**Harri Valpola**                                     *harri@system2ai.com*
**Alexander Ilin**                                    *alexilin@system2ai.com*
*System 2 AI*

**Reviewed on OpenReview:** *https://openreview.net/forum?id=drYpdSnRJk*

## Abstract

In many practical applications, large language models (LLMs) need to acquire new knowledge not present in their pre-training data. Efficiently leveraging this knowledge usually relies on supervised fine-tuning or retrieval-augmented generation (RAG). Although RAG has emerged as the industry standard for knowledge injection, fine-tuning has not yet achieved comparable success. This paper proposes utilizing prompt distillation, a self-distillation-based method previously explored primarily for style alignment and instruction tuning, to internalize new factual knowledge from free-form documents. Unlike prior methods, our approach requires neither larger teacher models nor structured knowledge formats. Across multiple LLM sizes and model families, we show that prompt distillation outperforms standard supervised fine-tuning and can even surpass RAG. We analyze the key factors contributing to prompt distillation's effectiveness and examine how it scales.[1]

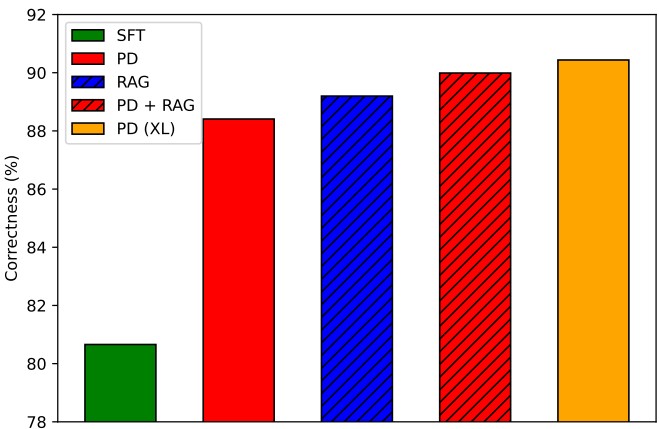

Figure 1: Average accuracy of responses across evaluation datasets. The proposed prompt distillation (PD) method performs competitively with RAG and surpasses supervised fine-tuning (SFT). Combining prompt distillation with RAG (PD+RAG) improves performance over standard RAG. Scaling up the fact coverage in PD training (PD XL) further enhances performance, outperforming all other methods.

---

[1]Code available at `https://github.com/kallekku/prompt-distillation`

# 1 Introduction

In many practical scenarios, large language models must integrate new, domain-specific factual knowledge absent from their original pre-training corpus. Two main strategies address this challenge: fine-tuning and retrieval-augmented generation (RAG). Fine-tuning integrates knowledge into model parameters but is sensitive to prompt variations and prone to overfitting. However, it helps reduce prompt length. In contrast, RAG excels at providing access to the most up-to-date information and is highly effective in responding to detailed queries, but requires an external knowledge base and can lead to very long prompts (Schick et al., 2024; Lewis et al., 2020).

RAG has become the industry standard for knowledge injection, demonstrating robust performance in various contexts (Chen et al., 2024). However, knowledge injection through supervised fine-tuning (SFT) has not yet achieved comparable success, as replicating RAG's performance has proven challenging (Ovadia et al., 2023; Mecklenburg et al., 2024). Recent approaches for injecting knowledge via SFT typically involve supervised learning using question-answers generated by prompting expert models like GPT-4 (Achiam et al., 2023) or Claude (Anthropic, 2024) using the new factual content. Since fine-tuning trains the student model to replicate expert answers exactly with answer tokens as targets, data quality is crucial. Consequently, using the best available model for data generation is common practice. Given that modern LLMs already encode vast amounts of knowledge, fine-tuning should, in principle, be capable of efficiently integrating new information, but current methods fall short. This paper seeks to close the performance gap between RAG and fine-tuning in knowledge injection tasks.

Classical fine-tuning has several challenges. First, using a different model for answer generation may introduce a mismatch between the expert and student answering styles (Gudibande et al., 2023). As a result, training may focus more on mimicking the answering style rather than learning the new factual information. Additionally, an intelligent expert may generate complex questions requiring high intelligence to answer, which the student may not possess. This issue was discussed, for example, by Mirzadeh et al. (2020) and Mitra et al. (2023), who carefully adjusted the prompts to allow a smaller model to learn more efficiently. Another concern arises from using tokens as targets since the same question can have multiple valid answers. Training a model to replicate the expert's answers verbatim risks overfitting, and the student model may not generalize well to answer new questions. This issue could be potentially mitigated by sampling multiple responses from the expert model. However, this makes the training data generation process much more expensive, particularly when using a large expert model or human-generated responses.

We explore *prompt distillation*, a self-distillation approach, to mitigate these limitations. Prompt distillation involves generating synthetic question-answer pairs about the new factual content using the LLM itself. It leverages the concept of self-distillation, where the student model learns from distributions over answer tokens produced by the teacher model, which additionally receives the new knowledge in its prompt (Fig. 2). By using the same model as both teacher and student (with a LoRA adapter (Hu et al., 2021)), we eliminate style and capacity mismatches. Consequently, training focuses specifically on learning factual knowledge, avoiding issues like verbatim memorization or unintended imitation of an expert model's answering style. No supervised annotations or structured formats are needed.

Prompt distillation has previously been explored primarily for stylistic alignment and instruction tuning. For example, Askell et al. (2021) used it for distilling alignment, Snell et al. (2022) for detailed task instructions, and Choi et al. (2022) for learning to act according to a persona. Qi et al. (2024) used a related approach for knowledge editing, relying on larger expert models and structured triplets. In contrast, our method applies self-distillation directly to unstructured, free-form documents without supervision, enabling efficient internalization of complex factual knowledge while remaining independent of larger models.

We conduct extensive evaluations with the Llama-3 (Dubey et al., 2024) and Qwen2.5 (Yang et al., 2024) model families on custom datasets derived from Squadshifts (Miller et al., 2020) and the multi-hop HotpotQA benchmark (Yang et al., 2018). Our findings show that prompt distillation significantly surpasses standard supervised fine-tuning for knowledge injection and reasoning. On Squadshifts, its closed-book performance can match that of open-book RAG. In the more complex HotpotQA, RAG is a strong baseline. However, PD improves closed-book performance substantially and, more importantly, enhances RAG when used together.

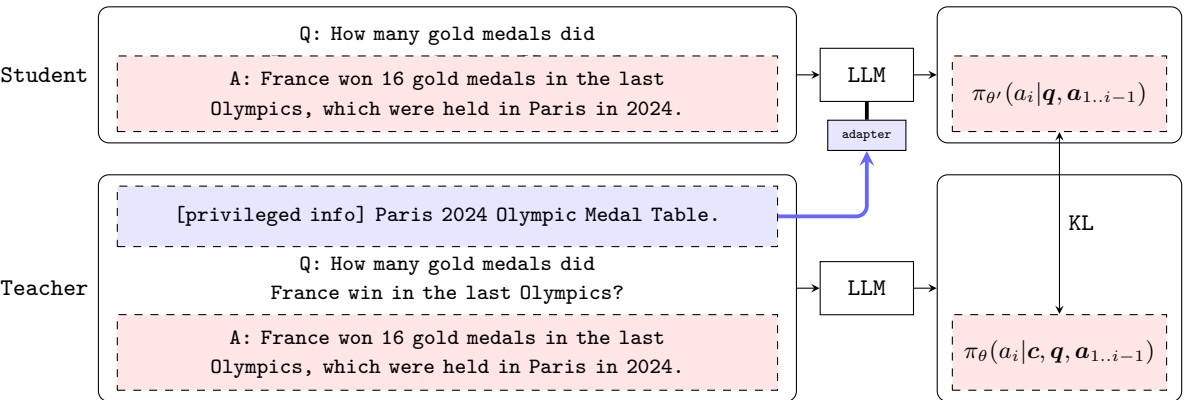

Figure 2: An overview of the prompt distillation approach. The privileged information $c$ in the teacher's prompt is distilled into the student's weights.

We summarize our main results from Squadshifts and compare prompt distillation to supervised fine-tuning and RAG in Figure 1.

This work highlights prompt distillation as a powerful yet under-explored method for injecting knowledge into LLMs without structured supervision or external retrieval. Our main contributions are: (1) introducing prompt distillation to internalize knowledge from free-form documents, (2) validating its effectiveness across multiple LLM families and sizes, and (3) analyzing key factors, including distillation and data generation temperature, the use of larger models as experts (for data generation) and teachers (providing target distributions), training data scale, and LoRA adapter size.

## 2 Prompt Distillation

Prompt distillation closely resembles imitation learning, where an expert model with additional knowledge $c$ generates demonstrations for the student to learn from. In knowledge injection, these demonstrations can be question-answer pairs $(q, a)$, both of which are multi-token sequences. Classical imitation learning (such as SFT for LLMs) restricts the student to learning from the expert's actions (tokens $a_i$) without accessing the expert's underlying policy. In contrast, prompt distillation uses an open-source teacher LLM to expose the complete policy $\pi_\theta(a_i|c, q, a_{1..i-1})$. By processing the entire sequence containing $c, q$, and $a$ in a single pass with causal masking, we obtain the teacher's logits at every token position, giving us the complete policy. This full policy from the teacher contains significantly richer information than isolated tokens alone. The goal of prompt distillation is thus to train a student model to approximate the teacher's full policy, but crucially, *without receiving context $c$ in its prompt*. Specifically, the student's policy should satisfy:

$$\pi_{\theta'}(a_i|q, a_{1..i-1}) \approx \pi_\theta(a_i|c, q, a_{1..i-1}) \iff D_{\mathrm{KL}}(\pi_\theta(a_i|c, q, a_{1..i-1}) \parallel \pi_{\theta'}(a_i|q, a_{1..i-1})) \approx 0$$

### 2.1 Data Generation

The first step of prompt distillation is the generation of expert demonstrations, which is a set of question-answer pairs $(q, a)$ about the new knowledge $c$. We first generate questions $q$ using an LLM with a high temperature ($\tau > 1$) to ensure that the questions are varied and that as few as possible are duplicated. Next, we generate answers $a$ using an LLM with a high temperature ($\tau > 1$) to encourage diversity. While this temperature is not high enough to produce frequent nonsense, it can occasionally introduce noisy or corrupted passages. This mild corruption does not harm training because the generated answers serve as inputs (not direct targets) for the student and teacher models (see Fig. 2). Crucially, the teacher's logits remain well-defined at each step, allowing the student to learn how to correct sub-optimal tokens if the teacher is capable of that. By distilling the teacher's policy $\pi_\theta(a_i|c, q, a_{1..i-1})$, the student learns to stay on the right track, which is the underlying idea of the DAGGER algorithm (Ross et al., 2011).

Since we want to copy the teacher's policy, we need to sample questions $q$ and answers $a$ where $c$ has a significant impact on the answer-generation policy $\pi_\theta(a_i|c, q, a_{1..i-1})$. Naturally, this impact is greatest when the questions and answers are related to $c$. However, having some questions and answers only distantly related to $c$ does not hurt; it merely reduces efficiency. In our experiments, we generate expert demonstrations in two steps: first, we generate the questions by either the model being fine-tuned or a specialized question-generating model (Nayak et al., 2024), and then we generate the answers either using the model itself or a larger model, to simulate the distillation of a more powerful expert.

## 2.2 Distillation

For the teacher model, we formulate a prompt consisting of the knowledge $c$, a question $q$, and an answer $a$. The teacher model computes logits $z_{i,v}$ for every position $i$ and token $v$ in the vocabulary $V$ in the answer $a$. The student model only receives the question $q$ and answer $a$ as input and computes the output logits $z'_{i,v}$. We perform knowledge transfer at a high temperature, following the classical knowledge distillation approach (Hinton et al., 2015). That is, the logits are converted into probabilities using the same temperature $T$ for both the teacher's and student's distributions:

$$\pi_\theta(v|c, q, a_{1..i-1}) = \frac{\exp(z_{i,v}/T)}{\sum_{v'} \exp(z_{i,v'}/T)}, \ \pi_{\theta'}(v|q, a_{1..i-1}) = \frac{\exp(z'_{i,v}/T)}{\sum_{v'} \exp(z'_{i,v'}/T)}$$

The loss is the average KL divergence between the two distributions for all answer positions $i = 1, \ldots, N_A$:

$$\mathcal{L} = \frac{1}{N_A} \sum_{i=1}^{N_A} D_{\mathrm{KL}}(i) = \frac{1}{N_A} \sum_{i=1}^{N_A} \sum_{v \in V} \pi_\theta(v|c, q, a_{1..i-1}) \log \frac{\pi_\theta(v|c, q, a_{1..i-1})}{\pi_{\theta'}(v|q, a_{1..i-1})}, \tag{1}$$

where $N_A$ is the number of answer tokens. The expected KL divergence between the teacher and student is equal to the mutual information between $c$ and $a_i$, where the expectation is taken over the distribution of $c$:

$$\mathbb{E}_c[D_{\mathrm{KL}}(\pi_\theta(a_i|c, q, a_{1..i-1}) \| \pi_{\theta'}(a_i|q, a_{1..i-1}))] = I(c; a_i \mid q, a_{1..i-1})$$

Mutual information measures the reduction in uncertainty of one random variable when the other is known. By aligning the student's distribution $\pi_{\theta'}(a_i|q, a_{1..i-1})$ with the teacher's, we minimize the *expected information gain* at inference time that would come from seeing $c$, ensuring the student internalizes the extra information from the context during training. A graphical illustration of the distillation is shown in Fig. 2.

We distill the knowledge at a high temperature $T > 1$, making the student focus more on less probable tokens. That is, the student learns to actively avoid the answers that the teacher is avoiding. To achieve the same effect using the cross-entropy loss with one-hot targets, one would need an enormous number of samples to realize that some answers must have a low probability. We briefly explored incorporating mid-layer activations in the loss but found that using the output logits alone was simpler and efficient enough.

## 2.3 Comparison to Fine-Tuning with Hard Targets

Cross-entropy loss with hard (one-hot) labels forces the student to assign all probability mass to a single token, potentially disrupting its existing distribution, even when it already aligns well with the teacher's knowledge. As a result, the student needs many training samples to average out these disturbances and approximate the desired distribution. Furthermore, these disturbances not only require the student to process many training samples to approximate the desired distribution but also increase catastrophic forgetting, as the forced alignment with one-hot labels can override previously learned knowledge (Kirkpatrick et al., 2017).

KL divergence loss with soft targets lets the student retain its distribution where the teacher adds no new information, leading to more efficient learning. In contrast, cross-entropy loss with hard labels treats the answer tokens generated by the expert as ground-truth targets. This requires sampling the answers at a low temperature to produce factually correct outputs. However, this can result in less diverse answers and a higher degree of overfitting to these specific targets during fine-tuning. Consequently, the fine-tuned student model may struggle to answer questions formulated differently.

## 2.4 Student Model

In our experiments, the student model is constructed to be the same as the teacher model, additionally equipped with a LoRA adapter (Hu et al., 2021):

$$\theta' = \theta + \Delta\theta \,,$$

where $\Delta\theta$ is LoRA initialized to $\Delta\theta = 0$. Thus, the two models are identical at the beginning of training. During training, the adapter's weights $\Delta\theta$ change as the new knowledge $\boldsymbol{c}$ gets distilled into the student's weights $\theta'$. The teacher remains unchanged. Our implementation uses the same network for both the student and the teacher, toggling the LoRA adapter to switch between roles. Thus, having two roles during training does not lead to increased memory consumption compared to SFT with a LoRA adapter. To save computing resources at the cost of storage, the teacher logits can be stored during answer generation to eliminate all additional compute during fine-tuning compared to SFT with hard targets. Alternatively, if multi-epoch training is performed, the logits can be computed and saved during the first epoch for subsequent epochs.

## 2.5 Regularization

Generally, we are not interested in only injecting knowledge into a model. We want the model to remain usable for other tasks and prevent overfitting. This can be easily achieved in prompt distillation: assume a dataset of instruction-response pairs, and in particular, an instruction $\boldsymbol{i}$ and a response $\boldsymbol{r}$. Assuming that the instruction-response pair is unrelated to all the injected knowledge $\mathcal{C}$, we want the student's response to remain unchanged during the fine-tuning. In practice, since domain-specific data usually differs from the general instruction tuning set, we treat these sets as disjoint; any rare overlap is negligible and can safely be ignored. We add the following KL divergence between the student and teacher to the loss function:

$$\mathcal{L}_{reg} = \frac{1}{N_R} \sum_{i=1}^{N_R} D_{\text{KL-reg}}(i) = \frac{1}{N_R} \sum_{i=1}^{N_R} \sum_{v \in V} \pi_\theta(v | \boldsymbol{i}, \boldsymbol{r}_{1..i-1}) \log \frac{\pi_\theta(v | \boldsymbol{i}, \boldsymbol{r}_{1..i-1})}{\pi_{\theta'}(v | \boldsymbol{i}, \boldsymbol{r}_{1..i-1})} \,, \tag{2}$$

where $N_R$ is the number of tokens in the response sequence. Minimizing this KL divergence forces the student's output distribution to remain close to its initial distribution for generic instructions, thus preventing catastrophic forgetting while allowing the model to internalize the new knowledge.

## 3 Related Work

**Prompt Engineering and In-Context Learning** enable users to guide model behavior through instructions and examples (Brown et al., 2020; Sanh et al., 2021; Lewis et al., 2020; Liu et al., 2023). Retrieval-augmented generation (RAG) extends this by integrating externally retrieved knowledge into prompts (Lewis et al., 2020; Izacard & Grave, 2020). However, longer, complex prompts make it harder for LLMs to process and integrate relevant information into their reasoning effectively. Prompt-based knowledge is transient, as it disappears once the prompt changes. Despite increasing context lengths, LLMs remain limited in managing extensive information within a single prompt (Liu et al., 2024b).

**Unsupervised Fine-Tuning (UFT)** of LLMs has been used for knowledge injection with mixed results. Training LLMs on next-token prediction tasks on new documents can lead to performance gains as large language models learn to memorize their training data (Gururangan et al., 2020; Carlini et al., 2021; 2022). However, UFT has often underperformed in knowledge injection compared to RAG, particularly in tasks involving new factual information. Ovadia et al. (2023) highlight LLMs' challenges in learning new facts through UFT, while Nayak et al. (2024) show it can sometimes reverse prior instruction tuning gains.

**Supervised Fine-Tuning** for knowledge injection proceeds by training pre-trained LLMs on tasks derived from new documents, either generated by standard LLMs (Mecklenburg et al., 2024) or specialized models (Nayak et al., 2024). RAFT (Zhang et al., 2024) integrates RAG with SFT to filter irrelevant information and prevent performance degradation due to distractors (Shi et al., 2023a; Mallen et al., 2022). Liu et al. (2024a) propose a two-stage approach consisting of continual pre-training and SFT. Gupta et al. (2024)

show that fine-tuning and RAG can be combined to maximize domain-specific performance. Unlike RAG, fine-tuning can enable cumulative knowledge integration without increasing prompt length, allowing models to improve on complex tasks (Snell et al., 2022; Choi et al., 2022; Ye et al., 2023; Liu et al., 2024a).

**Knowledge Distillation** is a technique where a smaller model learns from a larger model, first introduced by Hinton et al. (2015). It has been widely applied in NLP (see, e.g., Sanh et al., 2019). In the context of LLMs, distillation has been used to transfer the abilities of larger, proprietary LLMs to smaller models or otherwise improve the performance of smaller models with the larger teacher model (Peng et al., 2023; Taori et al., 2023; Chiang et al., 2023; Xu et al., 2023a;b; Mukherjee et al., 2023). Recent advances refine teacher prompts to elicit more detailed responses, including reasoning explanations (Mukherjee et al., 2023). Mitra et al. (2023) tuned the teacher's prompt to better align with the student's learning needs. Research by Wang et al. (2020) and Mukherjee & Awadallah (2020) suggests that utilizing richer signals, such as logits, intermediate representations, and attention states, can enhance the distillation process.

**Self-Distillation** allows a model to learn from its own outputs, performing self-improvement. The teacher can access privileged information, such as cleaner inputs, while the student model learns from noisier or incomplete data. This technique has been particularly effective in semi-supervised learning (see, e.g., Tarvainen & Valpola, 2017 and He et al. (2019)). Self-distillation is particularly relevant to our work, as it eliminates the need for a more advanced teacher model, simplifying the distillation while eliminating the initial mismatch between the teacher and the student.

**Context Distillation** has been explored in prior work, including Askell et al. (2021), which distilled human-aligned conversational examples into model weights to improve answering style rather than factual knowledge, and Snell et al. (2022), which demonstrated distillation of detailed task instruction and reasoning steps. Choi et al. (2022) distilled brief instructions and persona-defining prompts and trained a separate model to generate distillation data, whereas we show that standard LLM prompting alone suffices for learning, achieving competitive performance with RAG. Qi et al. (2024) applied context distillation for knowledge editing, representing the knowledge to be edited as structured triplets and leveraging large expert models. In contrast, we focus on a different problem, knowledge injection from unstructured documents, and employ self-distillation, making our approach independent of larger models while often achieving superior performance.

## 4 Experiments

We evaluate prompt distillation for knowledge injection using a dataset derived from Squadshifts (Miller et al., 2020). The original dataset was designed for an *open-book* setting, where the model retrieves answers from a provided document. The dataset consists of a passage of text, a question, and spans of text within the passage that contain the answer. We fine-tune models for a *closed-book* setting, where answers must be generated without the document in the prompt. Since many original questions lack sufficient context to be understood without the document, we reformulate them while preserving the original answers. We use Llama-3-70B-Instruct to generate reformulated questions, which constitute our test set for evaluating fine-tuned models.

Table 1: Summary of the dataset derived from Squadshifts

| Dataset | Documents | Number of Tokens |
|---------|-----------|------------------|
| Amazon | 207 | 363,031 |
| New Wiki | 203 | 252,035 |
| NYT | 188 | 338,918 |
| Reddit | 209 | 368,214 |

The test set includes 1,000 questions from each Squadshifts variant: Wikipedia, New York Times articles, Reddit posts, and Amazon product reviews. The number of passages used corresponds to the documents for the first 1,000 questions, ranging from 188 (NYT) to 209 (Reddit) (see Table 1). We perform experiments on the four individual subsets separately. To ensure a valid evaluation, test questions must probe knowledge not already known to the base model. To test this, we evaluate the performances of the base models on the test questions (see the base model results in Table 2). All prompts used for question generation, answer generation, and grading are presented in Appendix O.

### 4.1 Implementation of Prompt Distillation

We evaluate prompt distillation on three instruct-tuned models from different families and sizes: Llama-3-8B, Qwen2.5-14B, and Qwen2.5-3B. We generate 30 training questions for each test question using the evaluated model, sampling at a high temperature (1.5) to ensure diversity. Training questions are generated solely from the source documents without access to test questions. We do not explicitly prevent training questions from resembling test questions, but any similarity is incidental, as test questions are reformulations (produced by a different model) of the originals, which the training question generator never accessed. For each evaluated model, we generate answers to the training questions using the model itself with temperature $\tau = 1.5$.

We use the training questions to fine-tune each model using the prompt distillation approach. The student model uses a LoRA adapter, with rank 1024 for the 3B and 8B models and 512 for the 14B model, applied to all layers. We train all models using AdamW with a learning rate of $10^{-5}$, linear LR warmup, and a batch size of 4 per GPU. We fine-tune the 8B model on one AMD MI250X GPU for 24 hours ($\approx 10$ epochs). The 3B model is trained on one GPU and the 14B model on 8 GPUs for five epochs. In initial experiments, we exclude regularization due to its added computational cost. At test time, we present each test question individually to the fine-tuned model, sampling an answer with a temperature of 0.25. For the complete set of hyperparameters for prompt distillation, please see Table 8 in Appendix D.

### 4.2 Answer Grading

We use LLM-as-a-judge grading (Zheng et al., 2023), relying on Llama-3-8B-Instruct with a Chain-of-Thought-like prompt. The grading prompt follows two steps: first, the model generates a justification for the grade; then, it assigns the grade itself. Grades are binary: true or false. This two-step prompt significantly improves grading accuracy, as it encourages the model to reason before deciding, similar to a chain-of-thought process. Without explicit reasoning, the model struggles to produce reliable judgments. Manual inspection of a test subset estimated that approximately 2 % of grades were clearly incorrect (see Appendix L). To further validate grading accuracy, we re-evaluated selected experiments using Qwen2.5-32B-Instruct (see Appendix M). The choice of grader had minimal impact, except in the case of non-fine-tuned instruct models without RAG. The conclusions remain unchanged. Alongside LLM-as-a-judge, we used sub-string match grading, following Mallen et al. (2022). Here, an answer is marked as correct if any of the gold answers appear as a substring in the generated response after simple normalization. The detailed results and a discussion of this metric are presented in Appendix N. This alternative metric confirmed the same relative performance trends observed with our primary LLM-based evaluation.

### 4.3 Baselines

**Supervised Fine-Tuning (SFT).** We use the same 30 training questions as in prompt distillation. However, for SFT, the answers are re-sampled at a lower temperature (0.25) to reduce variance in responses. We fine-tune the models using a LoRA adapter and optimize the standard cross-entropy token loss. All other training parameters, including duration, infrastructure, and learning rate, remain consistent with prompt distillation. We fine-tune models using both their own generated answers and those from larger models, simulating API-based distillation without access to logits.

**RAG.** We test the performance of the instruct models with RAG. We employ two different retrieval techniques: BM25 and embedding-based. For BM25, we create a document database by tokenizing all context paragraphs from the evaluation dataset with the Llama-3 and Qwen tokenizers. We then tokenize the questions using the same tokenizer and use the Okapi BM25 ranking function (Robertson et al., 1995) from the rank-bm25 Python library to perform retrieval. In the embedding-based approach, we embed all the documents and questions with OpenAI's Embedding-API, using the *text-embedding-3-small* model. We use cosine similarity to retrieve the most relevant documents. For RAG, we retrieve the $k = 7$ most relevant documents per query and append them to the prompt. The value of $k$ is maximized within Llama-3-8B-Instruct's context limits. In our preliminary tests, we found that the performance increases with a larger number of documents in the prompt. We further tested RAG in combination with fine-tuned models.

Table 2: The average answer correctness (%) on the question answering task in the closed-book (upper part) and RAG (lower part) scenarios. The uncertainty is two standard errors of the mean.

**Base Model: Llama-3-8B-Instruct**

| Method | Amazon | New Wiki | NYT | Reddit |
|---|---|---|---|---|
| Prompt Distillation | **86.1** ±0.2 | **94.4** ±0.3 | **93.6** ±0.6 | **79.5** ±1.4 |
| Supervised Fine-Tuning | 75.9 ±1.8 | 89.5 ±0.2 | 87.5 ±0.2 | 69.8 ±0.4 |
| Unsupervised Fine-Tuning | 39.5 ±1.6 | 63.1 ±0.6 | 52.6 ±1.4 | 30.9 ±2.7 |
| Llama-3-8B-Instruct | 22.1 | 61.2 | 38.2 | 20.8 |
| Prompt Distillation + RAG | **88.5** ±0.3 | **96.7** ±0.2 | **96.9** ±0.2 | 77.8 ±1.6 |
| SFT w/ Distractors + RAG | 87.0 ±0.7 | 95.1 ±0.4 | 94.2 ±0.1 | **82.4** ±0.8 |
| Llama-3-8B-Instruct + RAG | 86.3 | 95.6 | 96.3 | 78.6 |

**Base Model: Qwen2.5-14B-Instruct**

| Method | Amazon | New Wiki | NYT | Reddit |
|---|---|---|---|---|
| Prompt Distillation | **85.5** ±0.6 | **94.9** ±0.3 | **92.6** ±0.8 | **77.6** ±0.4 |
| Supervised Fine-Tuning | 78.2 ±1.7 | 90.2 ±0.1 | 87.1 ±0.6 | 71.5 ±0.8 |
| Qwen2.5-14B-Instruct | 16.5 | 61.9 | 31.2 | 16.7 |
| Prompt Distillation + RAG | **88.4** ±0.4 | **97.3** ±0.1 | **96.3** ±0.4 | **83.6** ±1.3 |
| SFT + RAG | 87.9 ±0.3 | 96.6 ±0.6 | 95.7 ±0.6 | 83.1 ±0.3 |
| Qwen2.5-14B-Instruct + RAG | 87.1 | 97.1 | 94.8 | 81.3 |

**Base Model: Qwen2.5-3B-Instruct**

| Method | Amazon | New Wiki | NYT | Reddit |
|---|---|---|---|---|
| Prompt Distillation | **76.4** ±1.5 | **90.0** ±1.3 | **84.6** ±0.3 | **65.6** ±0.6 |
| Supervised Fine-Tuning | 69.1 ±0.7 | 84.3 ±1.4 | 76.6 ±0.8 | 59.8 ±0.5 |
| Qwen2.5-3B-Instruct | 12.6 | 46.7 | 18.6 | 12.1 |
| Prompt Distillation + RAG | **86.5** ±0.6 | **94.8** ±0.6 | **95.0** ±0.3 | **75.6** ±1.0 |
| SFT + RAG | 81.7 ±0.8 | 92.2 ±0.9 | 89.6 ±1.0 | 68.7 ±0.9 |
| Qwen2.5-3B-Instruct + RAG | 85.4 | 94.5 | 91.9 | 75.3 |

**Supervised Fine-Tuning with Distractors**. We implement an SFT method inspired by RAFT (Zhang et al., 2024). During fine-tuning, two random distractor documents from the same domain are appended to the context alongside the golden document. The order of the documents is randomized to prevent the model from identifying the correct document based solely on position. To improve robustness, the correct snippet is omitted with a 40% probability. This ensures the model learns to answer questions even when the retrieved context lacks the correct document, aligning with the optimal setting in Zhang et al. 2024 (see their Fig. 5). This approach combines elements of both open-book and closed-book fine-tuning. Training data is generated using a Chain-of-Thoughts prompt, as emphasized in the original work.

**Unsupervised Fine-Tuning**. This setting mirrors the base model pre-training, aiming to predict the next token. The paragraphs are sampled from the documents in the Squadshifts datasets by chunking (splitting into smaller snippets) while keeping overlap, ensuring context from one snippet carries into the next. Three epochs provided the best performance after testing different training durations and learning rates.

All fine-tuned models use the same training questions, LoRA adapter, and training durations to match the PD setup, except unsupervised fine-tuning, where longer training causes overfitting and no training questions are utilized. Retrieval methods (Okapi BM25, OpenAI Embeddings) and retrieved snippet count remained constant across methods. Experiments used three seeds, except for the base LLM, which was not re-trained.

Table 3: The average answer correctness (%) of prompt distillation (PD) and SFT with different-sized Llama-3 (8B & 70B) and Qwen2.5 (3B, 14B & 72B) models generating the answer tokens (expert) and target distributions (teacher). The top row for each subtable is the standard PD configuration. The uncertainty is two standard errors of the mean.

**Base Model: Llama-3-8B-Instruct**

| Method | Expert | Teacher | Amazon | New Wiki | NYT | Reddit |
|--------|--------|---------|--------|----------|-----|--------|
| PD | 8B | 8B | **86.1** ±0.2 | **94.4** ±0.3 | **93.6** ±0.6 | 79.5 ±1.4 |
| PD | 70B | 8B | 84.8 ±0.4 | 94.3 ±0.2 | 93.2 ±0.2 | 78.4 ±0.2 |
| PD | 70B | 70B | 84.0 ±0.5 | 93.0 ±0.5 | 92.3 ±0.1 | **82.0** ±0.3 |
| PD | 72B (Qwen2.5) | 8B | 84.8 ±1.5 | 93.2 ±0.3 | 92.3 ±0.5 | 78.2 ±1.2 |
| SFT | 8B | – | 75.9 ±1.8 | 89.5 ±0.2 | 87.5 ±0.2 | 69.8 ±0.4 |
| SFT | 70B | – | 75.6 ±1.0 | 89.2 ±0.9 | 85.6 ±1.2 | 70.9 ±0.3 |
| SFT | 72B (Qwen2.5) | – | 74.2 ±0.5 | 88.7 ±1.2 | 84.0 ±0.6 | 60.5 ±2.3 |

**Base Model: Qwen2.5-14B-Instruct**

| Method | Expert | Teacher | Amazon | New Wiki | NYT | Reddit |
|--------|--------|---------|--------|----------|-----|--------|
| PD | 14B | 14B | **85.5** ±0.6 | **94.9** ±0.3 | **92.6** ±0.8 | **77.6** ±0.4 |
| PD | 72B | 14B | 85.1 ±0.5 | 94.3 ±0.4 | 92.4 ±0.5 | 76.6 ±0.9 |
| PD | 72B | 72B | 81.7 ±0.6 | 93.6 ±0.6 | 90.8 ±0.6 | 69.9 ±0.2 |
| SFT | 14B | – | 78.2 ±1.7 | 90.2 ±0.1 | 87.1 ±0.6 | 71.5 ±0.8 |
| SFT | 72B | – | 76.8 ±0.9 | 90.0 ±0.5 | 86.3 ±1.0 | 68.7 ±0.5 |

**Base Model: Qwen2.5-3B-Instruct**

| Method | Expert | Teacher | Amazon | New Wiki | NYT | Reddit |
|--------|--------|---------|--------|----------|-----|--------|
| PD | 3B | 3B | 76.4 ±1.5 | 90.0 ±1.3 | 84.6 ±0.3 | 65.6 ±0.6 |
| PD | 14B | 3B | **78.4** ±0.6 | **90.8** ±0.2 | **87.3** ±0.6 | **69.5** ±0.6 |
| SFT | 3B | – | 69.1 ±0.7 | 84.3 ±1.4 | 76.6 ±0.8 | 59.8 ±0.5 |
| SFT | 14B | – | 74.3 ±1.0 | 87.6 ±1.2 | 85.3 ±1.7 | 67.7 ±0.6 |

## 4.4 Results

**Results without Larger Models**. In Table 2, we evaluate the performance of Prompt Distillation (PD) using three different base models, Llama-3-8B-Instruct, Qwen2.5-3B-Instruct, and Qwen2.5-14B-Instruct and compare PD to the baselines under two settings: a plain question-answering scenario in a closed-book setting (upper part of each table) and question-answering enhanced by RAG (lower part). For Llama-3, PD is the strongest method among the four non-RAG approaches, significantly outperforming supervised fine-tuning and achieving near-RAG performance. In contrast, unsupervised fine-tuning performs poorly. With RAG, PD achieves the highest overall performance. Notably, despite no explicit fine-tuning for RAG, PD effectively leverages retrieved context without disregarding it.

SFT with distractors achieves the best RAG performance on the Reddit dataset. Retrieving and attending to the correct document on Reddit proved particularly difficult, as many posts follow similar patterns, often seeking help or advice, making it easy to confuse relevant and irrelevant passages when answering questions. Adding distractor documents during supervised fine-tuning enhances robustness to misleading retrieved context but reduces closed-book performance without RAG. This is an orthogonal direction to PD and could potentially be combined for an even higher potential performance gain in the RAG setting.

With the Qwen2.5-family models, we compare PD to SFT, the strongest baseline with Llama-3-8B-Instruct. We also compare PD to the instruct model, both without RAG and with OpenAI Embeddings-based RAG, as it outperformed BM25 with token-based embeddings. For Qwen2.5-14B, PD outperforms SFT with and without RAG and improves on the instruct model when combined with RAG. Across all datasets, the performance gap between PD and instruct RAG is smaller than between SFT and PD. The same conclusions

Table 4: The average answer correctness (%) of prompt distillation and supervised fine-tuning with different temperatures, when the training data has been generated by Llama-3-8B Instruct. The top row is the standard prompt distillation configuration. The uncertainty is two standard errors of the mean.

| Method | T | $\tau$ | Amazon | New Wiki | NYT | Reddit |
|--------|-----|------|------------------|------------------|------------------|------------------|
| PD | 2 | 1.5 | **86.1** ±0.2 | **94.4** ±0.3 | **93.6** ±0.6 | **79.5** ±1.4 |
| PD | 1 | 1.5 | 83.8 ±0.9 | 93.3 ±0.5 | 91.8 ±0.2 | 78.2 ±0.9 |
| PD | 2 | 0.25 | 83.6 ±1.1 | 93.9 ±0.1 | 92.7 ±0.5 | 79.0 ±1.6 |
| SFT | N/A | 0.25 | 75.9 ±1.8 | 89.5 ±0.2 | 87.5 ±0.2 | 69.8 ±0.4 |
| SFT | N/A | 1.5 | 75.0 ±0.2 | 88.8 ±0.8 | 85.1 ±1.1 | 64.1 ±1.9 |

hold for the 3B model, where PD remains the strongest method in both settings, consistently surpassing the non-fine-tuned RAG baseline.

Almost all differences between PD and baseline methods exceed the two standard error confidence intervals. Qualitative examples of PD with Llama-3-8B-Instruct and Qwen2.5-14B-Instruct are provided in Appendices P and Q. Appendix E presents complete RAG results, comparing BM25 and OpenAI's Embedding-API. We also include Oracle retrieval results to quantify the performance loss from imperfect retrieval.

**Distillation from Larger Models**. Traditional distillation is commonly used to transfer knowledge and reasoning abilities from a larger model to a smaller one. In contrast, PD employs *self-distillation*, where the student model begins as a copy of the instruct model, also acting as both the expert (generating QA pairs from the context $c$) and the teacher (producing token-distribution targets for the student). During training, the expert and teacher remain unchanged while the student learns by updating its LoRA adapter, gradually diverging from its initial state. Generally, the expert and teacher models can be chosen independently, except that logit-based distillation requires a shared vocabulary between the student and teacher.

We systematically evaluate how changing the expert or teacher model affects PD and SFT performance (Table 3). SFT has no teacher, only the expert that generates target answers, equivalent to the standard distillation approach via closed APIs. For Llama-3-8B-Instruct, *self-distillation*, that is, using the same model to generate both data and logits (expert = teacher = 8B), yields superior knowledge injection compared to using a larger Llama-3-70B-Instruct or Qwen2.5-72B-Instruct as either the expert or teacher. Similarly, for Qwen2.5-14B-Instruct, self-distillation again proves more effective than relying on data/logits from a 72B model in the same family, likely because style mismatches overshadow any benefits of the more powerful model. An exception arises for the smaller model, Qwen2.5-3B, which struggles to generate high-quality training data. Using a larger expert with the 3B model as its own teacher improves results here, but even self-distillation still outperforms standard SFT. Because Qwen2.5-3B has a different vocabulary from its larger siblings, logit-based distillation from the bigger models is not directly possible.

Our findings suggest that mismatches in answering style often outweigh the benefits of a more powerful expert or teacher in knowledge injection. Especially under parameter-efficient fine-tuning, a larger teacher may inadvertently push the student to mimic the teacher's style rather than absorb new facts. Hence, self-distillation, where student = teacher = expert, often leads to better knowledge retention.

### 4.5 Ablations and Scaling Behavior

In this section, we present further experiments to analyze prompt distillation with Llama-3-8B-Instruct. We provide detailed results from these ablations in Appendices F-I. Additionally, in Appendix J, we explore reverse KL divergence as a loss function, and in Appendix K, we analyze why using larger expert models does not always enhance performance.

**Temperature**. In Table 4, we show the importance of using higher sampling temperatures for training data generation ($\tau$) and during distillation ($T$). Reducing the value of either temperature leads to a degradation in the PD performance. This highlights the importance of maximizing data coverage, actively learning to avoid incorrect answers, and leveraging the flatter probability distribution produced by the higher temperature,

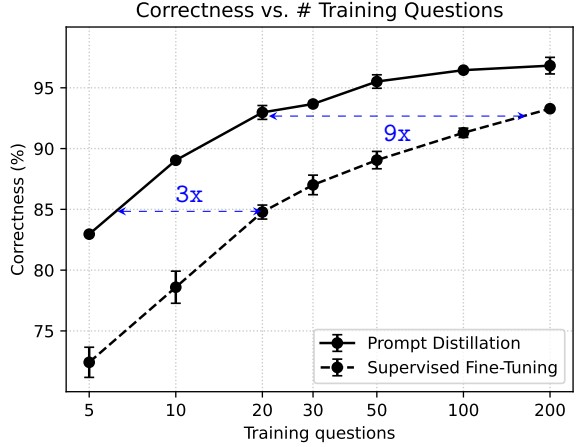

Figure 3: Average correctness (%) of PD and SFT *vs.* the number of questions per test question.

Figure 4: Answer correctness (%) of PD with Bonito vs. Llama-3-8B-Instruct for question generation.

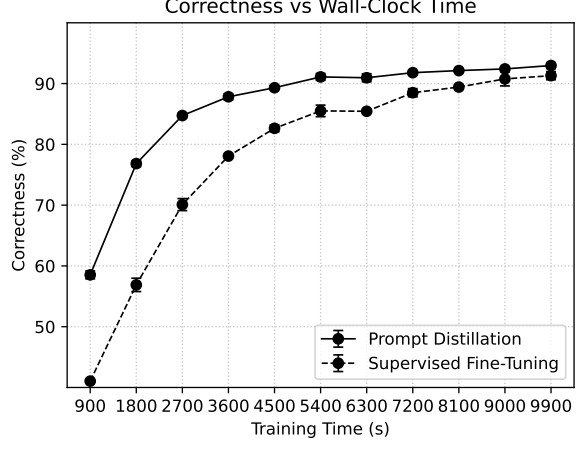
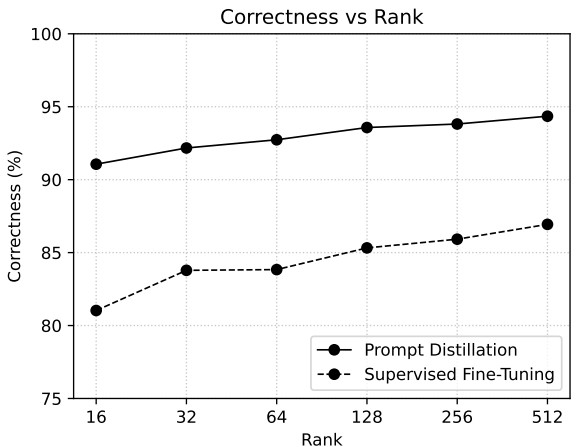

Figure 5: Answer correctness (%) of PD and SFT *vs.* training time (seconds).

Figure 6: Answer correctness (%) of PD and SFT *vs.* LoRA adapter rank.

which encourages the model to pay increased attention to the probabilities of alternative tokens. In contrast, SFT uses the expert answers directly as the ground truth. While higher temperatures improve data coverage, they also introduce lower-quality answers, ultimately degrading SFT performance.

**Number of Training Questions**. In Figure 3 and Appendix F, we study the impact of the number of training questions on the performance of PD and SFT on Llama-3-8B-Instruct. These results highlight that PD is substantially more data-efficient than SFT, with the efficiency gap widening as the number of training questions increases. To match the performance of PD with 20 training questions, SFT needs an order of magnitude more training data. In our experiments, SFT did not reach the asymptotic performance of PD.

**Question-Generating Model**. In Figure 4, we analyze the role of the training question-generating model by comparing few-shot prompting of Llama-3-8B-Instruct to our reproduction of Bonito (Nayak et al., 2024). This model has been fine-tuned to generate zero-shot questions given the document. We hypothesize that Bonito's slight performance advantage at larger training question counts stems from improved fact coverage. However, the benefit is minor. For smaller data budgets, few-shot prompting is sufficient.

**Wall-Clock Time**. In the simplest PD implementation, each gradient step requires two forward passes: one without the LoRA adapter (to retrieve teacher logits) and one with it (for the student). In contrast, SFT

Table 5: The average answer correctness (%) on the question answering task with **Llama-3-8B-Instruct** and the Tülu 3 dataset used to prevent catastrophic forgetting. The uncertainty is two standard errors.

| Method | Amazon | New Wiki | NYT | Reddit |
|---|---|---|---|---|
| Prompt Distillation | 86.1 ±0.2 | 94.4 ±0.3 | 93.6 ±0.6 | 79.5 ±1.4 |
| Prompt Distillation + Tülu 3 | 84.9 ±0.4 | 94.5 ±1.1 | 92.9 ±0.3 | 79.0 ±0.3 |
| Prompt Distillation XL + Tülu 3 | **89.4** ±0.5 | **96.2** ±0.4 | 95.9 ±0.4 | **80.3** ±0.8 |
| Llama-3-8B-Instruct + RAG | 86.3 | 95.6 | **96.3** | 78.6 |

Table 6: The average answer correctness (%) of prompt distillation and supervised fine-tuning on MMLU-Pro. The mean is computed over 6 subjects. PD suffers less from forgetting than SFT.

| Method | Engineering | History | Law | Math | Physics | Psychology | Mean |
|---|---|---|---|---|---|---|---|
| Prompt Distillation | 27.8 | 36.7 | 23.4 | 33.0 | 29.2 | 53.9 | 34.0 |
| Prompt Distillation + Tülu 3 | 31.1 | 41.0 | 25.8 | 35.1 | 32.3 | 57.8 | 37.2 |
| Prompt Distillation XL + Tülu 3 | **31.6** | 41.2 | **26.8** | 35.5 | 34.3 | 57.9 | 37.9 |
| Supervised Fine-Tuning | 28.4 | 32.6 | 22.6 | 30.3 | 26.9 | 50.7 | 31.9 |
| Llama-3-8B-Instruct | 31.3 | **42.3** | 26.5 | **36.1** | **34.4** | **59.4** | **38.3** |
| Llama-3-8B | 25.5 | 36.2 | 19.6 | 30.4 | 31.4 | 53.3 | 32.7 |

does not require a teacher pass. To analyze the impact on wall-clock time, we trained Llama-3-8B-Instruct, saving model snapshots every 900 seconds and evaluating performance. To isolate efficiency differences rather than asymptotic performance, we used 30 training questions for PD and 200 for SFT, leading to a similar final performance (see Fig. 3). The results (Fig. 5) show that PD's higher data efficiency compensates for its additional forward pass, making it more time-efficient than SFT. Note that PD with self-distillation can be further optimized by pre-computing and storing teacher logits during answer generation.

**LoRA Rank**. We used large LoRA adapter sizes in most experiments to ensure that model capacity was not a limiting factor. However, in real-world applications, memory constraints may require smaller adapters. To assess this impact, we varied the adapter rank for both PD and SFT while keeping LoRA alpha equal to twice the rank. The results (Fig. 6) show that both methods scale predictably, with performance increasing monotonically as the number of learnable parameters increases. PD consistently outperforms SFT at every rank, demonstrating that it is also a more parameter-efficient approach to knowledge injection.

### 4.6 Prompt Distillation with Regularization

Finally, we investigate the efficacy of PD training in mitigating catastrophic forgetting when augmented with regularization. We use the Tülu 3 SFT Dataset (Lambert et al., 2024) as a regularization mechanism, with Llama-3-8B-Instruct as the student model. Mini-batches contain an equal proportion (50:50) of samples from the PD and Tülu 3 datasets. We applied the standard KL divergence loss function for items from the PD dataset (Eq. 1). In contrast, the regularization loss (Eq. 2) was utilized for Tülu dataset instances where no additional context $c$ was present. This loss trains the student and teacher outputs to match.

We compared the trained model to standard PD and SFT across our evaluation datasets, with MMLU-Pro (Wang et al., 2024b) serving as a proxy metric for general model performance. The empirical results (Tables 5 and 6) show that incorporating the Tülu 3 dataset slightly reduces performance in knowledge injection tasks but significantly improves resistance to catastrophic forgetting on MMLU-Pro. This approach narrows the gap to Llama-3-8B-Instruct to approximately one percentage point, demonstrating strong retention of general capabilities. Standard PD is still superior to the base Llama-3-8B model. In contrast, SFT on the low-temperature PD dataset significantly degraded instruction tuning benefits on MMLU-Pro, causing performance to drop below the base model's. These results suggest that using soft labels in PD mitigates performance degradation compared to SFT with one-hot targets.

Table 7: The average answer correctness (%) on HotpotQA across all methods and base models, in the closed-book (top) and RAG (bottom) settings. Uncertainty is two standard errors of the mean.

| Method | Llama-3-8B-Instruct | | Qwen2.5-14B-Instruct | | Qwen2.5-3B-Instruct | |
|---|---|---|---|---|---|---|
| PD (30 Qs) | 70.0 | ±0.8 | 71.1 | ±2.5 | 57.9 | ±0.9 |
| PD (100/50 Qs) | **73.7** | ±0.1 | **73.4** | ±0.2 | **59.1** | ±2.5 |
| SFT (30 Qs) | 63.3 | ±2.9 | 66.8 | ±0.4 | 56.4 | ±0.9 |
| SFT (100/50 Qs) | 68.9 | ±2.7 | 68.8 | ±2.2 | 51.8 | ±2.9 |
| Base Model | 53.0 | | 43.5 | | 28.8 | |
| PD (30 Qs) + RAG | 81.7 | ±0.8 | 86.0 | ±1.4 | 76.4 | ±1.9 |
| PD (100/50 Qs) + RAG | **82.1** | ±1.7 | **87.0** | ±0.3 | **76.9** | ±1.4 |
| SFT (30 Qs) + RAG | 74.8 | ±1.8 | 80.2 | ±4.1 | 75.1 | ±1.0 |
| SFT (100/50 Qs) + RAG | 79.6 | ±1.3 | 78.5 | ±12.1 | 75.2 | ±0.2 |
| Base Model + RAG | 79.5 | | 83.5 | | 73.5 | |

To improve performance across evaluation datasets and MMLU-Pro, we introduced the following three modifications to PD to form PD XL: (1) increasing the number of training questions from 30 to 200 per test question, (2) replacing Llama-3-8B-Instruct with our reproduction of Bonito for question generation, and (3) limiting training to a single epoch. Then, we combined PD XL with regularization based on the Tülu 3 dataset, leading to the Prompt Distillation XL + Tülu 3 variant, which we evaluated with Llama-3-8B-Instruct as the base model. This optimized approach reduces the performance gap to less than 0.5 percentage points relative to Llama-3-8B-Instruct on MMLU-Pro while exceeding the OpenAI Embeddings-based RAG baseline on knowledge injection tasks. Incorporating preference fine-tuning and RLVR, key post-SFT steps in the Tülu 3 pipeline (Lambert et al., 2024), could potentially further close the gap.

### 4.7 Experiments on HotpotQA

While our evaluations on Squadshifts demonstrate PD's capacity to learn and retain factual knowledge, many real-world applications require models to reason using this incorporated knowledge to answer more complex questions. To assess this capability, we evaluate our method on HotpotQA (Yang et al., 2018), a multi-hop QA benchmark designed to test reasoning. To answer the test questions successfully, the model has to synthesize information from multiple supporting documents to arrive at the correct answer.

We use the first 1,000 questions from the validation set of the HotpotQA distractor setting for our experiments. Each test question is typically associated with ten context paragraphs: two containing the necessary supporting facts and eight being distractors. Our objective is to inject the knowledge from these ten paragraphs into the student's LoRA adapter. This allows the student model to perform multi-hop reasoning to answer the test questions in a closed-book setting. We generate single-hop training questions based on each provided paragraph separately to ensure the model can learn the factual content from each of the context paragraphs. The test questions, on the other hand, require multi-hop reasoning. Thus, we are testing PD in an out-of-distribution setting where test questions are more complex than the training questions. However, while our current question generation pipeline focuses on single-document contexts for simplicity and to specifically test the generalization from standard QA to multi-hop reasoning, there is no inherent limitation to PD that would prevent it from being trained directly for multi-hop reasoning. This could involve developing a pipeline to select multiple related documents and generate training questions that explicitly require synthesizing information, similar to how the HotpotQA dataset has been constructed. However, such an extension is beyond the scope of the current work, where our focus is on evaluating PD for general knowledge injection.

We follow the self-distillation principle from the Squadshifts experiments. An expert model generates the QA pairs, a teacher generates the logits, and a student equipped with a LoRA adapter is trained. All three

use the same model. For PD, we sample the expert-generated answers at a temperature of 1.5. We use SFT as a baseline and generate the answers at a temperature of 0.25, also aligning with Squadshifts.

We evaluate two main training data configurations for HotpotQA. In the first setting, we generated 30 training questions from each of the ten context paragraphs associated with a HotpotQA test question. We train both Llama-3-8B-Instruct and the Qwen models for five epochs. Then, to explore the impact of greater data density per paragraph, which might benefit the multi-hop task, we increased the number of training questions to 100 per paragraph for Llama-3-8B-Instruct and 50 per paragraph for the Qwen-based models. We used a smaller number for Qwen models, as their outputs were generally more verbose. In this configuration, the Llama-3-8B-Instruct model was trained for two epochs, and the Qwen models were trained for three. These settings were chosen based on preliminary experiments to optimize performance within our computational budget.

We evaluate performance in two settings: a closed-book scenario without RAG and using RAG with OpenAI Embeddings (specifically, *text-embedding-3-small*). For RAG, we retrieved the top $k = 10$ most relevant paragraphs from the pool of all paragraphs associated with our 1,000 test questions. The value of $k$ was chosen to align with the standard HotpotQA setup, where 10 context paragraphs are used in conjunction with each question. Answer grading was performed by Llama-3-8B-Instruct using the same Chain-of-Thought prompting methodology as used for Squadshifts. For completeness, results using substring match grading, following Mallen et al. (2022), are provided in Appendix N.

The results are presented in Table 7. They demonstrate that Prompt Distillation consistently outperforms both SFT and the base models across all tested architectures, data configurations, and evaluation settings on HotpotQA. Increasing the number of training questions per paragraph from 30 to 100/50 generally leads to improved performance for PD, which highlights its ability to utilize more extensive training data effectively. Our results suggest that PD not only effectively injects new knowledge (as shown with Squadshifts) but also enhances the model's ability to utilize this newly acquired knowledge in more complex, multi-step reasoning tasks. When combined with RAG, models trained with PD generally show superior performance, which indicates that the model can better combine the injected and retrieved information for multi-hop reasoning. Our results suggest that PD enables a deeper assimilation of knowledge rather than superficial recall, which allows PD to apply the knowledge in novel and more demanding settings.

## 5 Conclusion

In this paper, we propose using prompt distillation (PD) to inject knowledge from free-form documents into LLMs and analyze factors impacting its performance. Our method uses self-distillation, where a student model learns to replicate the answers of a teacher model with access to the target knowledge in its prompt. We show that PD outperforms traditional supervised fine-tuning (SFT) across model sizes and families, achieving higher accuracy with less data while being more efficient in terms of wall-clock time and parameters. Unlike SFT, PD does not depend on larger teacher models, provided the base model is sufficiently capable. In knowledge injection tasks from Squadshifts, PD achieves closed-book performance competitive with RAG when the number of training questions and the LoRA rank are sufficient. To the best of our knowledge, this has not been achieved with SFT. Furthermore, on the complex multi-hop reasoning benchmark HotpotQA, PD substantially improves the base model's performance and, notably, can enhance the performance of RAG systems when used in conjunction.

We believe storing knowledge within model weights offers a scalable alternative to RAG systems, which rely on long prompts and external databases, making them prone to retrieval errors. Effective knowledge internalization requires integrating new information with prior knowledge. PD can facilitate this more naturally than structured knowledge bases, which require ongoing maintenance. PD could be especially useful for domains like corporate knowledge management, where information is largely unstructured. One of the main challenges in this study was the absence of standardized benchmarks for unsupervised knowledge injection. We addressed this by modifying Squadshifts data to ensure that questions remained understandable without their original context, making comparing RAG and fine-tuning more fair. This modification improved the performance of RAG-based methods through improved retrieval quality. The modified questions have been included in the supplementary material. For evaluation, we primarily relied on LLM-as-a-judge methods

using models from different families (Llama & Qwen), verifying that grading outputs were consistent and rankings remained unaffected. Furthermore, we used substring match grading, which generally yielded similar conclusions despite variances in absolute scores. We also manually verified a subset of the LLM grades to ensure reliability. Finally, if the base model is too small, using a more powerful model as the expert can improve performance.

Our knowledge injection technique could aid the development of agentic LLMs, potentially enabling self-improving models that distill new insights directly into their weights. Future research could explore hybrid approaches combining PD with RAG and investigate how distractor documents impact model robustness. A particularly promising future work direction is optimizing and diversifying training data generation strategies for Prompt Distillation. For instance, optimizing question generation to maximize fact coverage or developing methods for multi-document training questions could further improve PD in factual recall and multi-hop reasoning, respectively. More generally, investigating how tailored synthetic data can lead to even greater performance for PD in a wide range of downstream tasks could further extend PD's capabilities.

### Acknowledgments

This work was supported by the Research Council of Finland (Flagship programme: Finnish Center for Artificial Intelligence FCAI, and grants 352986, 358246) and EU (H2020 grant 101016775 and NextGenerationEU). We thank Joni Pajarinen for helpful discussions and Minttu Alakuijala for contributions to the codebase. We thank CSC - IT Center for Science in Finland for providing access to the LUMI supercomputer, and the Aalto Science-IT project for additional computational resources.

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

## A    Reproducibility

The codebase for implementing prompt distillation, the evaluation questions, and instructions for replication of the main results can be found in the following GitHub repository: `https://github.com/kallekku/prompt-distillation`.

## B    Related Work: Retrieval-Augmented Language Models

Retrieval-Augmented Language Models integrate retrieval directly into the language model architecture (Guu et al., 2020; Borgeaud et al., 2022; Khandelwal et al., 2019; Izacard et al., 2023; Huang et al., 2023). Large autoregressive transformers can be retrofitted with retrieval capabilities during a fine-tuning or a continual pre-training phase (Lin et al., 2023; Liu et al., 2024c; Wang et al., 2023), or they can be converted into explicit

retrievers for RAG (Ma et al., 2024; Xu et al., 2024b). The LLM can also be combined with a retriever without fine-tuning (Shi et al., 2023b; Ram et al., 2023). Methods to compress the retrieved context for more effective inference (Xu et al., 2024a; Zhou et al., 2024; Wang et al., 2024a; Chevalier et al., 2023) are related to our work, as we aim to compress context directly into model weights through prompt distillation, reducing the need for complex retrieval pipelines and making inference cheaper.

## C   Related Work: More Detailed Review of Context Distillation

In prior work, context distillation has been used for in-context learning and qualitatively modifying LLM behavior. The idea of distilling context into model weights was used by Askell et al. (2021) to improve LLM alignment with human values to produce polite, helpful, and accurate answers. They distill a prompt that contains fourteen human-assistant conversations in which the assistant always follows the desired policy. Snell et al. (2022) demonstrated the distillation of more detailed task instructions, in-context examples, and step-by-step reasoning. Choi et al. (2022) used the distillation idea to distill short instruction prompts that, for example, define a persona participating in a conversation or contain task instructions. They additionally trained a model to generate input data for the distillation process, using examples from the considered benchmarks as training data.

Qi et al. (2024) proposed using a distillation approach to edit knowledge retained in the LLM weights. The new factual knowledge is assumed to exist as triplets (entity, relation, object). The triplets are converted into short statements by large proprietary LLMs, which are used as additional context for the teacher's prompt. Compared to that paper, we neither assume the existence of new knowledge in the triplet form nor data generation by larger LLMs. Instead, we learn new facts from raw documents without any supervision.

# D   Hyperparameters

Table 8: Hyperparameters used in our experiments. We either performed a grid search to determine the optimal values or selected the largest feasible values that fit within our computational budget and GPU memory.

| Hyperparameter | Value |
|---|---|
| Temperature for question generation (Llama & Qwen) | 1.5 |
| Temperature for question generation (Bonito) | 1.25 |
| Temperature for teacher answer generation (PD) | 1.5 |
| Temperature for teacher answer generation (SFT) | 0.25 |
| Maximum teacher answer length, PD dataset (tokens) | 512 |
| Maximum total prompt length, PD dataset (tokens) | 768 |
| Maximum total prompt length, Tülu 3 dataset (tokens) | 1024 |
| Temperature for sampling evaluation answers | 0.25 |
| Maximum answer length, evaluation | 500 |
| Number of documents to be retrieved | 7 |
| Learning rate | 1e-5 |
| Temperature for KL divergence loss | 2.0 |
| Batch size | 4 |
| Batch size (Low-rank experiments & PD XL & Qwen2.5-14B-Instruct) | 32 |
| Gradient accumulation steps | 1 |
| Adam weight decay | 0.1 |
| Maximum gradient norm | 1.0 |
| Precision | BF16 |
| GPU | AMD Instinct™ MI250X |
| LoRA rank (3B, 8B) | 1024 |
| LoRA rank (14B) | 512 |
| LoRA type | Full |
| LoRA alpha | 2 * rank |
| Learning rate warmup duration (Llama-3) | ≈ 1 epoch |
| Learning rate warmup duration (Qwen2.5) | 100 steps |
| Learning rate warmup type | Linear |
| Training duration (Llama-3) | ≈ 10 epochs |
| Training duration (Qwen2.5) | 5 epochs |
| Training duration (PD XL) | 1 epoch |
| Training duration (UFT) | 3 epochs |
| Learning rate for UFT | 5e-5 |

# E    Results Obtained with RAG

Table 9: The average answer correctness (%) of prompt distillation and baseline methods in the RAG-based setting with **Llama-3-8B-Instruct**. Retrieval was performed using BM25 and OpenAI Embeddings with cosine similarity. The uncertainty is two standard errors of the mean.

| Method | RAG | Amazon | New Wiki | NYT | Reddit |
|---|---|---|---|---|---|
| Prompt Distillation | BM25 | 87.0 ±0.4 | 96.4 ±0.6 | 96.7 ±0.1 | 70.5 ±2.7 |
| Prompt Distillation | emb | **88.5** ±0.3 | **96.7** ±0.2 | **96.9** ±0.2 | 77.8 ±1.6 |
| Supervised Fine-Tuning | BM25 | 85.4 ±0.1 | 95.6 ±0.5 | 96.0 ±0.3 | 71.5 ±2.5 |
| Supervised Fine-Tuning | emb | 87.3 ±0.6 | 95.8 ±0.2 | 96.2 ±0.3 | 75.2 ±2.9 |
| Unsupervised Fine-Tuning | BM25 | 67.7 ±1.1 | 87.5 ±1.8 | 85.8 ±0.6 | 54.1 ±2.7 |
| Unsupervised Fine-Tuning | emb | 68.4 ±1.3 | 87.0 ±1.8 | 87.1 ±1.8 | 57.7 ±1.7 |
| SFT w/ Distractors | BM25 | 84.0 ±0.6 | 95.6 ±0.4 | 94.5 ±0.5 | 77.9 ±0.4 |
| SFT w/ Distractors | emb | 87.0 ±0.7 | 95.1 ±0.4 | 94.2 ±0.1 | **82.4** ±0.8 |
| Llama-3-8B-Instruct | BM25 | 82.4 | 94.4 | 94.8 | 69.2 |
| Llama-3-8B-Instruct | emb | 86.3 | 95.6 | 96.3 | 78.6 |
| Llama-3-8B-Instruct | Oracle | 94.7 | 98.4 | 98.1 | 84.3 |

Table 9 presents the results of prompt distillation (PD) and baseline methods in a retrieval-augmented generation (RAG) setting. We compare two retrieval strategies: BM25 and OpenAI Embeddings (denoted as "emb"). PD outperforms standard supervised fine-tuning (SFT) across all datasets and achieves the highest accuracy with OpenAI Embeddings-based retrieval. Adding distractors to SFT improves its robustness in the RAG setting on the Reddit dataset. The table also includes results for unsupervised fine-tuning with RAG, which underperforms compared to other methods. As an upper bound, we report results using an "Oracle" retrieval method, where the correct document is always retrieved. This highlights the impact of retrieval quality, with all methods falling short of Oracle's performance due to imperfect retrieval.

# F   Ablation: Impact of the Number of Training Questions

Table 10: The average answer correctness (%) of prompt distillation with a varying number of training questions generated by Bonito per test question on the New York Times dataset with **Llama-3-8B-Instruct**. Retrieval was performed using BM25 and OpenAI Embeddings with cosine similarity. The uncertainty is two standard errors of the mean.

| Method | 5 | 10 | 20 | 30 | 50 | 100 | 200 |
|---|---|---|---|---|---|---|---|
| PD | **83.0** ±0.0 | **89.0** ±0.1 | **93.0** ±0.6 | **93.7** ±0.2 | **95.5** ±0.6 | **96.5** ±0.1 | **96.8** ±0.7 |
| SFT | 72.4 ±1.2 | 78.6 ±1.3 | 84.8 ±0.6 | 87.0 ±0.8 | 89.1 ±0.7 | 91.3 ±0.4 | 93.3 ±0.1 |
| PD + BM25 | **95.2** ±0.4 | **95.3** ±0.2 | **96.2** ±0.2 | **96.2** ±0.3 | **96.6** ±0.2 | **96.6** ±0.1 | **96.2** ±0.1 |
| SFT + BM25 | 94.1 ±0.6 | 94.4 ±1.1 | 94.9 ±0.2 | 95.2 ±0.5 | 95.3 ±0.3 | 95.7 ±0.1 | 95.5 ±0.2 |
| PD + emb | **95.8** ±0.5 | **96.2** ±0.2 | **97.0** ±0.5 | **96.9** ±0.4 | **96.9** ±0.2 | **97.0** ±0.3 | **96.9** ±0.7 |
| SFT + emb | 94.9 ±0.3 | 95.0 ±0.4 | 95.4 ±0.4 | 96.3 ±0.1 | 95.4 ±0.4 | 96.1 ±0.5 | 96.6 ±0.5 |

Table 10 compares the sensitivity of prompt distillation and supervised fine-tuning with token loss to the number of questions per document in the training data with Llama-3-8B-Instruct. We sampled the questions using Bonito (Nayak et al., 2024) in this experiment. We found Bonito capable of generating competitive questions for the New York Times dataset. Increasing the number of questions provides a statistically significant benefit, particularly in the closed-book setting. For instance, the performance on NYT increases from 83.0% with five questions to 93.7% with 30 questions.

We observe that prompt distillation is much more sample-efficient than supervised fine-tuning with token loss, reaching similar performance with much fewer samples. We also see that the performance of supervised fine-tuning saturates on a slightly lower level than prompt distillation in the RAG setting, showcasing that prompt distillation is superior at convergence. Finally, note that the baseline performance for Llama-3-8B-Instruct on NYT was 38.2%, showing that even a small number of questions gives significant benefits in terms of the correctness percentage.

# G  Ablation: Effect of Question-Generating Model

Table 11: The average answer correctness (%) of prompt distillation with Bonito and Llama-3-8B-Instruct used for generating the training question on the New York Times dataset, with **Llama-3-8B-Instruct** as the trained model. The upper part of the table corresponds to the closed book results, whereas the lower half consists of the results in the RAG setting. Retrieval was performed using OpenAI Embeddings with cosine similarity. The uncertainty is two standard errors of the mean.

| Method | 5 | 10 | 20 | 30 | 50 | 100 |
|---|---|---|---|---|---|---|
| Bonito | 83.0 ±0.0 | 89.0 ±0.1 | **93.0** ±0.6 | **93.7** ±0.2 | **95.5** ±0.6 | **96.5** ±0.1 |
| Llama-3-8B-Instruct | **84.6** ±0.7 | **89.8** ±0.5 | 92.3 ±0.2 | 93.6 ±0.6 | 94.9 ±0.1 | 96.0 ±0.3 |
| Bonito + emb | **95.8** ±0.5 | **96.2** ±0.2 | **97.0** ±0.5 | **96.9** ±0.4 | 96.9 ±0.2 | 97.0 ±0.3 |
| Llama-3-8B-Instruct + emb | 94.9 ±0.3 | 95.4 ±0.3 | 96.2 ±0.3 | **96.9** ±0.2 | **97.3** ±0.2 | **97.1** ±0.3 |

Table 11 compares the performance of prompt distillation (PD) when using Bonito versus Llama-3-8B-Instruct to generate training questions on the New York Times dataset. We evaluate correctness across different numbers of training questions per test question, with and without retrieval using OpenAI Embeddings. The results do not indicate any significant differences between the methods. As the number of training questions increases, Bonito exhibits a marginal advantage in closed-book settings. However, with RAG using OpenAI Embeddings, performance converges at a similar level for both question-generation approaches, suggesting that retrieval mitigates differences in training question diversity. These findings suggest that while a fine-tuned question-generation model may provide benefits at scale, few-shot prompting of a general-purpose model is a viable alternative for PD.

# H  Ablation: Training Time Efficiency

Table 12: The average answer correctness (%) of prompt distillation and supervised fine-tuning as a function of training time (in seconds) on the New York Times dataset with **Llama-3-8B-Instruct**. The uncertainty is two standard errors of the mean.

| Training Time (s) | PD | SFT |
|---|---|---|
| 900 | **58.5** ±0.7 | 41.0 ±0.2 |
| 1800 | **76.8** ±0.5 | 56.9 ±1.1 |
| 2700 | **84.7** ±0.3 | 70.1 ±1.0 |
| 3600 | **87.8** ±0.5 | 78.1 ±0.3 |
| 4500 | **89.3** ±0.3 | 82.6 ±0.6 |
| 5400 | **91.1** ±0.6 | 85.5 ±1.0 |
| 6300 | **91.0** ±0.7 | 85.4 ±0.3 |
| 7200 | **91.8** ±0.3 | 88.5 ±0.7 |
| 8100 | **92.1** ±0.1 | 89.4 ±0.4 |
| 9000 | **92.4** ±0.2 | 90.8 ±1.2 |
| 9900 | **93.0** ±0.5 | 91.3 ±0.7 |

Table 12 presents the relationship between training time and performance for Prompt Distillation (PD) and Supervised Fine-Tuning (SFT) on the New York Times dataset using Llama-3-8B-Instruct. To ensure comparable asymptotic performance, we used 30 training questions per test question for PD and 200 for SFT. The results show that PD consistently outperforms SFT at every time interval, achieving higher correctness scores with fewer training steps. For instance, PD reaches 90% correctness in approximately 5,000 seconds, whereas SFT takes nearly twice as long to approach similar performance. These findings highlight that PD's superior sample efficiency outweighs the additional computational cost of two forward passes—one for the student and another for the teacher. Due to storage constraints and slow file systems, we did not implement optimizations such as pre-computed logits.

# I  Ablation: Impact of LoRA Rank on Performance

Table 13: The average answer correctness (%) of prompt distillation and supervised fine-tuning as a function of LoRA rank on the New York Times dataset. The uncertainty is two standard errors of the mean.

| Method | 16 | 32 | 64 | 128 | 256 | 512 |
|--------|------|------|------|------|------|------|
| SFT | 81.0 $\pm$1.0 | 83.8 $\pm$0.6 | 83.8 $\pm$0.7 | 85.3 $\pm$0.1 | 85.9 $\pm$0.1 | 86.9 $\pm$0.1 |
| PD | **90.9** $\pm$0.0 | **91.5** $\pm$0.0 | **92.9** $\pm$0.0 | **93.7** $\pm$0.0 | **93.7** $\pm$0.0 | **94.7** $\pm$0.0 |

Table 13 presents the average answer correctness for prompt distillation (PD) and supervised fine-tuning (SFT) as a function of LoRA adapter rank on the New York Times dataset using Llama-3-8B-Instruct. We observe a clear trend where increasing the LoRA rank improves performance for both methods, with PD consistently outperforming SFT at all ranks. Even at the lowest rank (16), PD surpasses SFT at its highest, demonstrating PD's superior parameter efficiency in knowledge injection tasks.

## J  Ablation: Reverse KL Divergence

Table 14: The average answer correctness (%) of prompt distillation with different KL divergences and temperatures with **Llama-3-8B-Instruct**. The top row is the standard prompt distillation configuration. The uncertainty is two standard errors of the mean.

| KL | T | $\tau$ | Amazon | New Wiki | NYT | Reddit |
|---|---|---|---|---|---|---|
| Forward | 2 | 1.5 | **86.1** ±0.2 | **94.4** ±0.3 | **93.6** ±0.6 | **79.5** ±1.4 |
| Reverse | 1 | 0.25 | 75.5 ±0.8 | 88.4 ±0.5 | 84.4 ±1.0 | 70.2 ±1.3 |
| Reverse | 1 | 1.5 | 77.3 ±0.7 | 89.8 ±0.2 | 86.7 ±0.4 | 71.4 ±2.2 |
| Reverse | 2 | 0.25 | 82.5 ±0.8 | 93.2 ±0.2 | 90.9 ±0.4 | 76.3 ±0.1 |
| Reverse | 2 | 1.5 | 84.1 ±0.5 | 93.4 ±0.7 | 92.1 ±0.3 | 76.0 ±0.6 |

Table 14 compares the performance of prompt distillation using forward KL divergence (standard configuration) versus reverse KL divergence at different temperatures. Forward KL consistently outperforms reverse KL across all datasets, demonstrating its superior effectiveness in knowledge injection. One intuitive reason is that forward KL divergence (KL(teacher ∥ student)) is mode covering, heavily penalizing the student for failing to capture the teacher's most probable tokens. By contrast, reverse KL divergence (KL(student ∥ teacher)) is mode seeking and does not penalize ignoring parts of the teacher's distribution as strongly. This disparity becomes especially problematic when the distillation temperature $T$ is lowered from two to one because the teacher's distribution becomes sharper (i.e., higher peaks on certain tokens). Under forward KL, the student is forced to learn these peaks more accurately (see Table 4), so the drop in performance is smaller. Under reverse KL, the student more readily collapses onto a few modes, struggling to match the teacher's distribution as a whole and leading to a significant drop in performance.

## K   Ablation: Counter-Intuitive Scaling Effects with Bigger Experts

Table 15: The average answer correctness (%) of prompt distillation on the New York Times dataset when training on different quintiles of answers generated by Qwen2.5-72B-Instruct. The training data is divided into five disjoint subsets based on either the initial Kullback-Leibler divergence (KLD) between the student and teacher or the teacher's entropy. The uncertainty is two standard errors of the mean.

| Type | Quintile | Amazon | New Wiki | NYT | Reddit |
|------|----------|--------|----------|-----|--------|
| Entropy | 1 | 67.0 ±1.2 | 75.0 ±0.5 | 71.1 ±0.3 | 66.9 ±0.2 |
| Entropy | 2 | 72.0 ±0.7 | **85.3** ±0.2 | 79.4 ±0.6 | **69.6** ±1.1 |
| Entropy | 3 | **73.2** ±0.4 | 84.5 ±0.1 | **81.8** ±1.0 | 65.9 ±2.3 |
| Entropy | 4 | 73.0 ±0.2 | 82.1 ±0.7 | 80.9 ±0.2 | 65.5 ±1.0 |
| Entropy | 5 | 69.1 ±1.9 | 77.0 ±0.2 | 76.4 ±1.0 | 57.2 ±1.0 |
| KLD | 1 | 63.8 ±1.3 | 70.6 ±1.5 | 66.8 ±1.4 | 50.2 ±1.2 |
| KLD | 2 | 71.3 ±0.6 | 79.5 ±1.3 | 77.6 ±1.0 | 59.7 ±0.4 |
| KLD | 3 | 72.8 ±1.7 | 82.0 ±0.8 | 82.1 ±0.3 | 66.6 ±0.5 |
| KLD | 4 | **73.6** ±0.4 | **86.8** ±0.6 | **84.1** ±0.4 | 67.9 ±0.8 |
| KLD | 5 | 69.5 ±0.4 | 84.1 ±1.3 | 79.8 ±0.4 | **68.8** ±0.8 |

Table 16: Comparison of teacher entropy and initial KL divergence between the teacher and student models for different data-generating models across datasets, with Llama-3-8B-Instruct as the student and teacher. In each block, the first row represents the standard prompt distillation setting, self-distillation.

| Expert | Amazon | New Wiki | NYT | Reddit | Mean |
|--------|--------|----------|-----|--------|------|
| **Teacher Entropy** | | | | | |
| Llama-3-8B-Instruct | 0.41 | 0.26 | 0.30 | 0.68 | 0.41 |
| Qwen2.5-72B-Instruct | 0.52 | 0.42 | 0.43 | 0.71 | 0.52 |
| **KL Divergence** | | | | | |
| Llama-3-8B-Instruct | 1.70 | 1.42 | 1.56 | 1.29 | 1.49 |
| Qwen2.5-72B-Instruct | 1.31 | 0.92 | 1.19 | 0.83 | 1.06 |

Our empirical results in Table 3 reveal an unexpected phenomenon: knowledge injection performance deteriorates when a larger, more capable model serves as the answer-generating expert. This observation contrasts with the conventional expectation that distillation from more powerful models consistently improves the performance of smaller models. While similar counter-intuitive effects have been noted in computer vision (Cho & Hariharan, 2019; Mirzadeh et al., 2020) and general reasoning tasks with LLMs (Gudibande et al., 2023; Mitra et al., 2023), our study extends these findings to the specific case of knowledge injection through fine-tuning.

To explore potential factors underlying this phenomenon, we examine two key statistical properties of the teacher model's outputs:

1. We hypothesize that **high entropy in the teacher's output distribution**, possibly due to task difficulty or stylistic divergence between the data-generating expert and the teacher, may lead to weaker or noisier training signals. Conversely, very low entropy might indicate that the base model already possesses the relevant knowledge, making additional knowledge transfer redundant

2. **The initial KL divergence between the student and teacher** distributions represents a rough upper bound on potential information gain during fine-tuning. A low divergence may imply that the student already knows the information or that the teacher's uncertain signal could limit learning.

We conducted experiments utilizing Llama-3-8B-Instruct as the base model and Qwen2.5-72B-Instruct as the answer-generating expert model. We partitioned the training corpus into quintiles based on entropy and KL divergence and independently fine-tuned Llama-3-8B-Instruct on each subset. As shown in Table 15, the results indicate that a higher initial KL divergence correlates with greater performance gains, although extremely large values can have adverse effects. Both extremes perform poorly for teacher entropy, and training on the second quintile in our setup yields the best results on average.

Comparative analysis of whole datasets (Table 16) generated by Llama-3-8B-Instruct and Qwen2.5-72B-Instruct reveals that when the training data is self-generated by the model (Llama), the teacher entropy is lower while the initial KL divergence is greater. This characteristic may help explain why Llama-3-8B-Instruct demonstrates superior performance as an expert compared to Qwen2.5-72B-Instruct (Table 3). However, we emphasize that these are correlations. Further study would be required to isolate and confirm the causal mechanisms, as there might be confounding factors.

## L    Manual Answer Grading

We performed a manual evaluation of the grading accuracy of Llama-3-8B-Instruct. We evaluated 1000 grades assigned by the grader across three models: our prompt distillation model, the SFT model, and the default Llama-3-8B-Instruct. This evaluation covered the RAG and closed-book settings for PD and SFT and the RAG setting for Llama-3-8B-Instruct. The proportion of clearly incorrect grades was low, at 2.2%, with over 90% of these clearly incorrect grades being false negatives, that is, instances where the grader marked correct responses as incorrect. The similarity in the proportion of incorrect grades across all models indicates no systematic bias favoring one model over another. Given the large volume of evaluations, manual grading would have been infeasible. We found Llama-3-8B-Instruct to be a good balance between speed and grading accuracy.

Table 17: Confusion matrix for grading performed by Llama-3-8B-Instruct for Token Loss and Prompt Distillation.

| Token Loss | Correct Label TRUE | Correct Label FALSE | Total |
|---|---|---|---|
| Graded TRUE | 310 | 1 | 311 |
| Graded FALSE | 5 | 84 | 89 |
| **Total** | 315 | 85 | 400 |

| Prompt Distillation | Correct Label TRUE | Correct Label FALSE | Total |
|---|---|---|---|
| Graded TRUE | 327 | 1 | 328 |
| Graded FALSE | 9 | 63 | 72 |
| **Total** | 336 | 64 | 400 |

# M Answer Grading with Qwen2.5-32B-Instruct

We re-graded the most critical experiments using Qwen2.5-32B-Instruct to verify the reliability of our grading approach. The results are presented in the tables included in this appendix. The re-evaluation with Qwen2.5-32B-Instruct demonstrated that the overall ranking of methods remained consistent across different grading models. While minor variations were observed, these differences were generally small, except for non-fine-tuned models without retrieval-augmented generation (RAG). In this setting, Qwen2.5-32B-Instruct applied stricter grading criteria than Llama-3-8B-Instruct, with a particularly pronounced gap when Llama-3-8B-Instruct graded its own outputs. However, the non-fine-tuned instruct model consistently exhibited the lowest performance. Consequently, the greater variance in grading does not affect the core conclusions of our study: (1) PD is superior to SFT across different settings, (2) PD combined with RAG outperforms the instruct model with RAG, and (3) the PD XL + Tülu 3 model surpasses the instruct model with RAG.

## M.1 Base Model: Llama-3-8B-Instruct

Table 18: Average answer correctness (%) on question answering task in the closed-book (upper part) and RAG (lower part) scenarios with **Llama-3-8B-Instruct**, when the grading has been done with Qwen2.5-32B-Instruct. The uncertainty is two standard errors of the mean.

| Method | Amazon | New Wiki | NYT | Reddit |
|---|---|---|---|---|
| Prompt Distillation | **85.6** ±0.3 | **92.9** ±0.5 | **92.9** ±0.3 | **80.3** ±0.5 |
| Supervised Fine-Tuning | 75.3 ±1.4 | 87.5 ±0.7 | 86.4 ±0.2 | 68.8 ±1.2 |
| Unsupervised Fine-Tuning | 39.6 ±2.7 | 58.3 ±1.0 | 50.5 ±1.1 | 29.1 ±2.7 |
| Base Model | 19.7 | 51.1 | 30.4 | 14.5 |
| Prompt Distillation + RAG | **88.9** ±0.4 | **96.0** ±0.2 | **96.7** ±0.7 | 80.7 ±1.3 |
| SFT w/ Distractors + RAG | 87.2 ±0.4 | 93.5 ±0.2 | 93.9 ±0.2 | **83.6** ±0.4 |
| Base Model + RAG | 85.3 | 94.9 | 95.6 | 78.4 |

Table 19: Difference in correctness depending on the grader (Qwen minus Llama) in percentage points.

| Method | Amazon | New Wiki | NYT | Reddit |
|---|---|---|---|---|
| Prompt Distillation | -0.5 | -1.5 | -0.7 | +0.8 |
| Supervised Fine-Tuning | -0.6 | -2.0 | -1.1 | -1.0 |
| Unsupervised Fine-Tuning | +0.1 | -4.8 | -2.1 | -1.8 |
| Base Model | -2.4 | -10.1 | -7.8 | -6.3 |
| Prompt Distillation + RAG | +0.4 | -0.8 | -0.3 | +2.9 |
| SFT w/ Distractors + RAG | +0.2 | -1.6 | -0.4 | +1.1 |
| Base Model + RAG | -1.0 | -0.7 | -0.7 | -0.2 |

### M.2 Base Model: Qwen2.5-14B-Instruct

Table 20: Average answer correctness (%) on question answering task in the closed-book (upper part) and RAG (lower part) scenarios with **Qwen2.5-14B-Instruct**, when the grading has been done with Qwen2.5-32B-Instruct. The uncertainty is two standard errors of the mean.

| Method | Amazon | New Wiki | NYT | Reddit |
|---|---|---|---|---|
| Prompt Distillation | **87.3** ±0.3 | **94.2** ±0.3 | **92.8** ±0.1 | **81.4** ±0.2 |
| Supervised Fine-Tuning | 79.8 ±1.2 | 88.5 ±1.1 | 87.0 ±1.0 | 72.3 ±0.8 |
| Qwen2.5-14B-Instruct | 17.7 | 56.4 | 27.7 | 13.5 |
| Prompt Distillation + RAG | **90.4** ±0.2 | **96.7** ±0.5 | **96.9** ±0.3 | **86.6** ±0.7 |
| SFT + RAG | 89.7 ±0.2 | 95.5 ±0.4 | 96.5 ±0.4 | 86.1 ±0.1 |
| Qwen2.5-14B-Instruct + RAG | 89.1 | **96.7** | 95.0 | 83.9 |

Table 21: Difference in correctness with **Qwen2.5-14B-Instruct** as the base model depending on the grader (Qwen minus Llama) in percentage points.

| Method | Amazon | New Wiki | NYT | Reddit |
|---|---|---|---|---|
| Prompt Distillation | +1.8 | -0.7 | +0.2 | +3.8 |
| Supervised Fine-Tuning | +1.6 | -1.7 | -0.2 | +0.8 |
| Base Model | +1.2 | -5.5 | -3.5 | -3.2 |
| Prompt Distillation + RAG | +2.0 | -0.6 | +0.6 | +3.0 |
| SFT + RAG | +1.8 | -1.0 | +0.8 | +3.0 |
| Base Model + RAG | +2.0 | -0.4 | +0.2 | +2.6 |

### M.3 Base Model: Qwen2.5-3B-Instruct

Table 22: Average answer correctness (%) on question answering task in the closed-book (upper part) and RAG (lower part) scenarios with **Qwen2.5-3B-Instruct**, when the grading has been done with Qwen2.5-32B-Instruct. The uncertainty is two standard errors of the mean.

| Method | Amazon | New Wiki | NYT | Reddit |
|---|---|---|---|---|
| Prompt Distillation | **76.8** ±0.3 | **86.6** ±1.1 | **84.4** ±0.5 | **64.9** ±1.6 |
| Supervised Fine-Tuning | 68.8 ±0.9 | 80.5 ±1.0 | 75.6 ±1.7 | 58.8 ±1.6 |
| Qwen2.5-3B-Instruct | 13.7 | 41.1 | 15.6 | 8.9 |
| Prompt Distillation + RAG | **87.0** ±0.4 | **93.5** ±0.5 | **94.5** ±0.1 | **78.0** ±1.2 |
| SFT + RAG | 82.1 ±0.9 | 90.5 ±1.2 | 90.4 ±0.1 | 72.7 ±0.4 |
| Qwen2.5-3B-Instruct + RAG | 85.5 | 92.2 | 91.7 | 76.9 |

Table 23: Difference in correctness with **Qwen2.5-3B-Instruct** as the base model depending on the grader (Qwen minus Llama) in percentage points.

| Method | Amazon | New Wiki | NYT | Reddit |
|---|---|---|---|---|
| Prompt Distillation | +0.4 | -3.4 | -0.2 | -0.7 |
| Supervised Fine-Tuning | -0.3 | -3.8 | -1.0 | -1.0 |
| Base Model | +1.1 | -5.6 | -3.0 | -3.2 |
| Prompt Distillation + RAG | +0.5 | -1.3 | -0.4 | +2.4 |
| SFT + RAG | +0.5 | -1.6 | +0.8 | +4.0 |
| Base Model + RAG | +0.1 | -2.3 | -0.2 | +1.6 |

### M.4 Prompt Distillation with Regularization

Table 24: Average answer correctness (%) on the question answering task with **Llama-3-8B-Instruct** and the Tülu 3 dataset used to prevent catastrophic forgetting and Qwen2.5-32B-Instruct as the grader. The uncertainty is two standard errors of the mean.

| Method | Amazon | New Wiki | NYT | Reddit |
|---|---|---|---|---|
| Prompt Distillation | 85.6 ±0.3 | 92.9 ±0.5 | 92.9 ±0.3 | 80.3 ±0.5 |
| Prompt Distillation + Tülu 3 | 85.0 ±0.4 | 93.0 ±0.1 | 92.5 ±0.2 | 80.0 ±0.3 |
| Prompt Distillation XL + Tülu 3 | **89.3** ±0.7 | **95.3** ±0.4 | **95.6** ±0.5 | **82.1** ±0.1 |
| Llama-3-8B-Instruct + RAG | 85.3 | 94.9 | **95.6** | 78.4 |

Table 25: Difference in correctness on the question answering task with **Llama-3-8B-Instruct** as the model and the Tülu 3 dataset used to prevent catastrophic forgetting, depending on the grader (Qwen minus Llama) in percentage points.

| Method | Amazon | New Wiki | NYT | Reddit |
|---|---|---|---|---|
| Prompt Distillation | -0.5 | -1.5 | -0.7 | +0.8 |
| Prompt Distillation + Tülu 3 | +0.0 | -1.5 | -0.4 | +0.9 |
| Prompt Distillation XL + Tülu 3 | -0.1 | -1.0 | -0.3 | +1.8 |
| Llama-3-8B-Instruct + RAG | -1.0 | -0.7 | -0.7 | -0.2 |

# N    Substring Match-Based Grading

We employed substring match-based grading as a complementary evaluation metric to our primary LLM-as-a-judge approach. This method, following Mallen et al. (2022), marks an answer as correct if any of the gold answers appear as a substring in the generated response after simple normalization (lowercasing, removal of punctuation, and the most common stopwords, including articles).

Our LLM-as-a-judge evaluations focus on semantic correctness. Substring match offers a stricter, lower-variance principle. Absolute scores from substring matches are lower than those from LLM-as-a-judge. For instance, with Llama-3-8B-Instruct, the average LLM-graded correctness for PD was 88.4%, while the average substring match was 69.1%. For PD+RAG, the LLM-graded correctness was 90.0%, with an average substring match of 78.1%. Hence, the mean grading gap is around 15%. This discrepancy is natural. Instruction-tuned models often generate more verbose answers than those typically found in datasets like Squadshifts. LLM judges recognize semantic equivalence, whereas substring match is still an inherently strict lexical metric (for instance, is the gold answer 6 or "six"). Furthermore, we maintain that RAG-based methods have an inherent advantage in lexical match evaluations because they can directly copy text from the provided context passages. Closed-book models might phrase the answer differently, even if they know it.

Overall, despite the differences in absolute scores, the crucial finding is that the relative performance ranking and the observed improvements of Prompt Distillation over baselines remain consistent across both substring match and LLM-as-a-judge evaluations, except that the gap between non-RAG and RAG-based methods is larger. Tables 26-30 present the detailed substring match correctness scores for our most important experiments.

Table 26: Average substring match correctness (%) on question answering task in the closed-book (upper part) and RAG (lower part) scenarios with **Llama-3-8B-Instruct**. The uncertainty is two standard errors of the mean.

| Base Model: Llama-3-8B-Instruct | | | | |
|---|---|---|---|---|
| Method | Amazon | New Wiki | NYT | Reddit |
| Prompt Distillation | **59.1** ±0.5 | **75.6** ±0.6 | **79.6** ±0.2 | **62.1** ±1.4 |
| Supervised Fine-Tuning | 51.8 ±0.6 | 71.0 ±1.0 | 74.5 ±0.5 | 53.4 ±0.5 |
| Unsupervised Fine-Tuning | 27.2 ±0.6 | 51.1 ±1.8 | 46.9 ±1.9 | 27.7 ±1.2 |
| Llama-3-8B-Instruct | 12.5 | 31.8 | 23.0 | 12.3 |
| Prompt Distillation + RAG | **69.9** ±0.5 | **86.1** ±0.3 | **87.7** ±0.1 | **68.6** ±0.5 |
| SFT w/ Distractors + RAG | 65.3 ±0.1 | 78.1 ±0.5 | 84.0 ±0.2 | 66.1 ±1.0 |
| Llama-3-8B-Instruct + RAG | 65.5 | 83.8 | 85.7 | 58.7 |

Table 27: Average substring match correctness (%) on question answering task in the closed-book (upper part) and RAG (lower part) scenarios with **Qwen2.5-14B-Instruct**. The uncertainty is two standard errors of the mean.

| Base Model: Qwen2.5-14B-Instruct | | | | |
|---|---|---|---|---|
| Method | Amazon | New Wiki | NYT | Reddit |
| Prompt Distillation | **60.0** ±0.7 | **72.8** ±0.5 | **79.0** ±0.4 | **62.8** ±0.6 |
| Supervised Fine-Tuning | 53.3 ±1.3 | 68.0 ±0.5 | 74.1 ±0.6 | 57.1 ±0.8 |
| Qwen2.5-14B-Instruct | 11.4 | 36.5 | 21.7 | 13.9 |
| Prompt Distillation + RAG | **67.0** ±0.5 | **80.3** ±0.4 | 84.5 ±0.1 | 67.4 ±0.7 |
| SFT + RAG | 66.9 ±0.9 | 79.9 ±0.3 | **84.7** ±0.4 | **68.0** ±0.3 |
| Qwen2.5-14B-Instruct + RAG | 65.5 | 77.2 | 82.9 | 66.2 |

Table 28: Average substring match correctness (%) on question answering task in the closed-book (upper part) and RAG (lower part) scenarios with **Qwen2.5-3B-Instruct**. The uncertainty is two standard errors of the mean.

| Base Model: Qwen2.5-3B-Instruct | | | | |
|---|---|---|---|---|
| Method | Amazon | New Wiki | NYT | Reddit |
| Prompt Distillation | **52.7** ±0.9 | **68.1** ±0.5 | **72.8** ±0.6 | **52.7** ±1.3 |
| Supervised Fine-Tuning | 48.5 ±1.4 | 65.8 ±0.4 | 66.1 ±0.5 | 50.2 ±1.6 |
| Qwen2.5-3B-Instruct | 10.3 | 28.9 | 14.4 | 11.3 |
| Prompt Distillation + RAG | **67.5** ±2.3 | **79.9** ±0.3 | **84.4** ±0.2 | **66.9** ±0.5 |
| SFT + RAG | 62.1 ±1.6 | 77.3 ±0.8 | 80.4 ±0.8 | 63.4 ±0.5 |
| Qwen2.5-3B-Instruct + RAG | 66.3 | 77.4 | 80.7 | 65.7 |

Table 29: Average substring match correctness (%) on the question answering task with **Llama-3-8B-Instruct** and the Tülu 3 dataset used to prevent catastrophic forgetting. The uncertainty is two standard errors of the mean.

| Method | Amazon | New Wiki | NYT | Reddit |
|---|---|---|---|---|
| Prompt Distillation | 59.1 ±0.5 | 75.6 ±0.6 | 79.6 ±0.2 | **64.0** ±0.9 |
| Prompt Distillation + Tülu 3 | 59.2 ±0.5 | 76.3 ±0.6 | 79.6 ±0.3 | 61.1 ±0.2 |
| Prompt Distillation XL + Tülu 3 | 63.5 ±0.3 | 79.2 ±0.8 | 83.7 ±0.5 | 62.9 ±0.6 |
| Llama-3-8B-Instruct + RAG | **65.5** | **83.8** | **85.7** | 58.7 |

Table 30: The average substring match correctness (%) on HotpotQA across all methods and base models, in the closed-book (top) and RAG (bottom) settings. Uncertainty is two standard errors of the mean.

| Method | Llama-3-8B-Instruct | Qwen2.5-14B-Instruct | Qwen2.5-3B-Instruct |
|---|---|---|---|
| PD (30 Qs) | 54.7 ±0.5 | 56.9 ±2.7 | 45.3 ±1.0 |
| PD (100/50 Qs) | **59.4** ±0.6 | **58.9** ±0.9 | **47.0** ±1.4 |
| SFT (30 Qs) | 50.1 ±2.9 | 52.9 ±0.7 | 44.7 ±0.4 |
| SFT (100/50 Qs) | 56.0 ±0.6 | 55.8 ±0.9 | 41.8 ±1.4 |
| Base Model | 35.8 | 35.6 | 23.7 |
| PD (30 Qs) + RAG | 70.1 ±0.2 | 76.9 ±0.4 | 67.1 ±1.0 |
| PD (100/50 Qs) + RAG | 70.9 ±0.3 | **77.0** ±0.2 | **68.3** ±0.8 |
| SFT (30 Qs) + RAG | 63.6 ±0.8 | 71.3 ±7.4 | 67.2 ±0.8 |
| SFT (100/50 Qs) + RAG | **72.4** ±1.5 | 68.8 ±14.8 | 66.3 ±2.0 |
| Base Model + RAG | 69.1 | 73.7 | 65.4 |

## O Prompts

### O.1 Training Question Generation

#### O.1.1 Llama & Qwen

```
Here is a paragraph of text:
{context}

Please generate challenging trivia questions, at most 5, based on this
    paragraph. Do not make the questions multiple-choice. Do not assume that
    the person answering the questions has access to the paragraph. The
    questions must be understandable without access to the text. Do not
    output anything except the questions and format your output as in the
    following example:
<question>What is the capital of Japan?</question>
<question>How many months are there in a year?</question>
<question>What was the first name of Reagan?</question>
<question>How many goals did Messi score during the calendar year
    2012</question>
<question>Where is the Santa Monica pier located?</question>
```

#### O.1.2 Bonito

```
<|tasktype|>
extractive question answering
<|context|>
{text}
<|task|>
```

### O.2 Expert Answer Generation

```
{context}

———

{question}
```

### O.3 Student Fine-tuning

#### O.3.1 Prompt Distillation

```
{question}
```

#### O.3.2 SFT with Distractors

```
{context 1}

{context 2}

{context 3}

Please answer the following question. Reason step by step.
———
```

```
{question}
```

## O.4    Evaluation

### O.4.1    Closed-book evaluation

```
{question}
```

### O.4.2    RAG-based evaluation

```
{context1}

{context2}
...
{contextN}

Question: {question}
```

## O.5    Answer Grading

```
Here is a question, the list of accepted ground−truth answers and the
    proposed answer. Please evaluate if the answer is true or false and
    return the reasoning and grade as xml. If the answer matches any of the
    ground−truth answers, the grade should be true. Example:

Example
Question: What was the punishment for Mattingly for not getting a haircut?
Ground−truth answer: ['fined and benched', 'benched', 'fined and benched']
Proposed answer: Mattingly was benched for 20 games as punishment for not
    getting a haircut.

Output:
<reasoning>The ground−truth answer is that Mattingly was fined and benched
    for not getting a haircut. However, simply answering that Mattingly was
    benched is also a valid answer. The proposed answer correctly mentions
    that Mattingly was benched for not getting a haircut</reasoning>
<grade>true</grad>

Example:
Question: For what was Mattingly once fined and benched for?
Ground−truth answer: ['not getting a haircut']
Proposed Answer: Mattingly was fined and benched for not playing in a game,
    which was likely a form of protest against the team's management or the
    way he was being treated.

Output:
<reasoning>According to the ground−truth answer, Mattingly was fined and
    benched for not getting a haircut. The proposed answer does not answer
    the question accurately, and mentions a protest as a reason. However,
    this is clearly different from the ground−truth answer.</reasoning>
<grade>false</grade>

Example:
```

```
Question: Which baseball teams was Mattingly apart of?
Ground−truth answer: ['Yankee', 'Yankees']
Proposed answer: Mattingly was a part of the New York Yankees and the Los
    Angeles Dodgers baseball teams.

Output:
<reasoning>The ground−truth answer indicates that Mattingly was a part of the
    "Yankees" team. The proposed answer states that he was a part of the New
    York Yankees, which matches one of the ground−truth answers. However, it
    also mentions the Los Angeles Dodgers, which is not included in the
    ground−truth answers. Since the additional information about the Dodgers
    does not directly contradict the truth of him being on the Yankees, the
    primary question's focus remains valid.</reasoning>
<grade>true</grade>

Please grade the following question using the xml−template given:
Question: {question}
Ground−truth answer: {ground−truth answer}
Proposed answer: {answer}
```

### O.6   Question Preparation

```
Here is a piece of text:
{context}

Here is a question related to the text:
{question}

Here is a list of valid ground−truth answers:
{gt_answer}

Please re−write the question such that it can be fully understood and it
    makes sense without access to the text. Output the new question inside
    <question> xml tags, like this:

<question>Rewritten question</question>
```

## P   Example Outputs on New York Times

### P.1   When did Faust become CEO of Eaton Vance?

The correct answer to the question is 2007

### P.1.1   Llama-3-8B-Instruct

```
I think there may be some confusion here! After conducting a thorough search,
    I couldn't find any information on a person named Faust becoming the CEO
    of Eaton Vance. Eaton Vance is a global investment management company,
    and its CEO is currently Thomas E. Faust Jr., who has been in the role
    since 2010. However, I couldn't find any information on a person with the
    same name becoming CEO in the past. If you have any further questions or
    clarification regarding this, please feel free to ask!
```

### P.1.2 Prompt Distillation

```
According to the text, Faust became the CEO of Eaton Vance in 2007.
```

### P.2 When did Essid propose his cabinet?

The correct answer to the question is January 23.

#### P.2.1 Llama-3-8B-Instruct

```
I apologize, but I couldn't find any information on an individual named Essid
    proposing a cabinet. It's possible that you may have misspelled the name
    or that Essid is not a well-known public figure. Could you provide more
    context or clarify who Essid is? I'll do my best to help you find the
    information you're looking for.
```

#### P.2.2 Prompt Distillation

```
According to the article, Prime Minister Habib Essid proposed his first
    cabinet on January 23.
```

### P.3 Who is the book Splendid Cities by?

The correct answer to the question is Rosie Goodwin and Alice Chadwick

#### P.3.1 Llama-3-8B-Instruct

```
"Splendid Cities" is a book written by Donald Miller. The book was published
    in 1996 and explores the concept of urban design and the importance of
    creating livable and sustainable cities.
```

#### P.3.2 Prompt Distillation

```
The book "Splendid Cities" is by Rosie Goodwin and Alice Chadwick.
```

### P.4 Where did Kim Ki-Bum post a floral pattern from Ms. Bedford's book?

The correct answer to the question is Instagram

#### P.4.1 Llama-3-8B-Instruct

```
I'm not familiar with the specific context or individuals you're referring
    to. Kim Ki-Bum is a South Korean singer, songwriter, and actor, best
    known as a member of the K-pop group SHINee. Ms. Bedford's book is not a
    well-known publication, and I couldn't find any information about Kim
    Ki-Bum posting a floral pattern from her book. Could you provide more
    context or clarify which book or platform you're referring to? I'll do my
    best to help you find the answer.
```

#### P.4.2 Prompt Distillation

```
According to the text, Kim Ki-Bum posted a delicately colored-in floral
    pattern from Ms. Basford's book on Instagram, where he has 1.8 million
    followers.
```

**P.5  How much did the FBI originally offer in reward for information about Mr. Levinson?**

The correct answer to the question is $1 million.

### P.5.1  Llama-3-8B-Instruct

```
The FBI originally offered a $5 million reward for information leading to the
    safe return of Robert Levinson, a former FBI and CIA agent who went
    missing in Iran in 2007.
```

### P.5.2  Prompt Distillation

```
According to the article, the FBI originally offered a $1 million reward for
    information about Mr. Levinson in 2012.
```

# Q  Examples of Prompt Distillation Failures with Qwen2.5-14B-Instruct

### Q.1  Which country's military is notorious for its extensive use of acronyms?

The correct answer in the dataset is Indonesian military. The model is defaulting to its base behavior, not having fully internalized the document we're interested in that highlights the Indonesian military's use of acronyms.

```
The military known for its extensive use of acronyms is the United States'
    military. The passage mentions that the U.S. military and government
    agencies are particularly prolific in creating acronyms, with examples
    like \"A.W.O.L.\" (Absent Without Official Leave) and \"M.I.A.\" (Missing
    In Action)
```

### Q.2  What geological feature was pushed beneath the Earth's surface by the formation of the Andes?

The correct answer is parts of the Sunsás Orogen. The model gives a generic, reasonable answer, but the injection of the specific knowledge from the documents has failed.

```
The geological feature that was pushed beneath the Earth's surface by the
    formation of the Andes is part of the ancient cratons to the east. These
    cratons, which are the stable and oldest parts of the continental crust,
    were subjected to deformation and subduction as the Andean orogeny
    progressed. This process involved the collision and compression of
    tectonic plates, leading to the uplift and formation of the mountain
    range we know today as the Andes.
```

### Q.3  Was French sugar more expensive or less expensive than British sugar before the French Revolution?

The correct answer is cheaper. The model has simply not learned the facts correctly, but resorts to generic reasoning that leads to an incorrect conclusion.

```
Before the French Revolution, French sugar was more expensive than British
    sugar. This is because the British had a significant advantage in the
    sugar trade due to their control over important islands like Jamaica,
    Trinidad, and Barbados, as well as the territory of British Guiana. These
    colonies allowed Britain to produce sugar at a lower cost compared to
    France, making British sugar cheaper on the international market.
```

**Q.4   Who was appointed as the first head of the Armenian Church?**

The correct answer is Gregory. The model is close and has knowledge of the Armenian church, but incorrectly recalls the facts.

```
According to the information provided, Bartholomew was appointed as the first
    head of the Armenian Church. The text mentions that Thaddeus and
    Bartholomew were the apostles who brought Christianity to Armenia, and it
    specifically states that Bartholomew founded a nunnery and placed a
    portrait of the Virgin Mary there, indicating his significant role in
    establishing the early Christian community in Armenia.
```

**Q.5   Who made the statement "Who knows? Life is so unpredictable"?**

The correct answer is Hillary Clinton. The model does not recall that, and instead hallucinates Ms. Chow as the answer.

```
The statement "Who knows? Life is so unpredictable" was made by Ms. Chow.
```

