# OpenReview forum: "Efficient Knowledge Injection in LLMs via Self-Distillation"
_TMLR — Accepted by TMLR_

### Review · Reviewer_67aR · 2025-04-21

**Summary Of Contributions:**

This paper proposed a new knowledge injectin methods by first let teacher model generating ~30 augmented Q&A containing desired knowledge and force the student model to do imitation learning. The author has conducted the experiments on Qwen and Llama models and achieves comparable performnace using RAG.

**Audience:**

Yes

**Claims And Evidence:**

Yes

**Requested Changes:**

I hope the author can address two cons I mentioned.

**Strengths And Weaknesses:**

Pro:
Knowledge injecting is an important topics and this paper provides a very practical solution with experimental support.

Cons:
There are two parts I feel not clear.
1. The author mentioned that “ generate 30 training questions for each test question ” without explicitly prevent the similarity. But given the number of generated question, I still feel it would be better to study the relationship between test and train. Moreover, it would be even better if the knowledge used in test set is a more complicated combination of injected knowledge in train set.
2.You said the unsupervised tuning is worse than supervised, I feel this is solely due to the way you perform downstream evaluation, which requires some alignment. It is actually counterintuitive because most people believe knowlege is injected during pretraining/midtraining stages. I think the comparison is unfair.

---

> ### Author Response · Authors · 2025-05-20
>
> We thank the reviewer for their feedback and for recognizing Prompt Distillation (PD) as a practical solution for knowledge injection.
>
> 1. W1: Relationship between training and test questions.
>
> The reviewer raises a valid concern about the potential similarity between the 30 training questions generated and the single test question. It's important to reiterate that the model generating the training QA pairs only sees the source document. It has no access to the original questions or our re-formulated test questions. If the document contains limited facts, the model might generate questions about the same facts. We view this not as a flaw but as indicative of adequate coverage of the document's knowledge, which is required for efficient knowledge injection.
>
> However, this naturally leads to the question of whether the model can generalize beyond simple fact retrieval. To address the generalization question and test more complex knowledge integration, we evaluated our method on HotpotQA, a multi-hop reasoning benchmark. Answering the test questions necessitates combining information from multiple paragraphs. We used a closed-book setup with 1000 questions, corresponding to approximately 10,000 documents (paragraphs). For PD and SFT, training data (QA pairs) was generated from individual documents. Therefore, the QA pairs cannot mirror the test questions. Our preliminary HotpotQA results with Llama-3-8B-Instruct are as follows:
>
> | Method                    | Score (%) |
> |---------------------------|-----------|
> | Llama-3 + Oracle RAG      | 87.1      |
> | Llama-3 + PD + RAG        | 82.1      |
> | Llama-3 + SFT + RAG       | 79.6      |
> | Llama-3 + RAG             | 79.5      |
> | Llama-3 + PD              | 73.7      |
> | Llama-3 + SFT             | 68.9      |
> | Llama-3-8B-Instruct       | 53.0      |
>
> The strong performance of PD on HotpotQA, where it outperforms SFT and enhances RAG, demonstrates that PD does not merely perform superficial memorization of QA pairs. PD can internalize the factual knowledge from the documents and apply that knowledge to novel, more complex tasks. This addresses the concern about testing more complicated combinations of injected knowledge. We plan to include these results and results with Qwen in the revised manuscript.
>
> 2. W2: Comparison between Unsupervised Fine-Tuning (UFT) and PD/SFT:
>
> We fully agree that an LLM's knowledge is typically acquired during pre-training (and potentially mid-training). However, our work focuses on the challenge of post-hoc knowledge injection: updating an already pre-trained and instruction-tuned model with new, domain-specific information, often under constraints of limited computational resources. This scenario is typical for users who want to adapt off-the-shelf LLMs. PD enables significant performance gains with just 24 hours of training on a single GPU. In contrast, full pre-training is often not feasible in such contexts.
>
> For UFT in our experiments, we continued training on the next-token prediction objective on the new documents. We hypothesize that its lower performance on the targeted QA tasks, compared to SFT/PD, comes from the less direct learning signal. SFT/PD explicitly trains the model to recall and apply facts in a QA format. Multiple generated questions provide varied contexts for each fact. When training the model with UFT, the model only sees raw facts in a single context. This might be equivalent to learning from a single QA pair.
>
> We use the QA format for evaluation because that's the primary way users interact with and assess the knowledge in LLMs. We believe our UFT setup is the standard way to perform continued pre-training for domain adaptation. However, as noted, it can harm the model's alignment. Achieving comparable performance to PD/SFT would require substantial additional effort (if even possible), such as further instruction tuning, which is what our experiments highlight. Our results should not be interpreted as evidence of unsupervised fine-tuning not being a generally valid strategy for acquiring knowledge during pre-training; we show that doing simple UFT post-training is insufficient for QA knowledge injection. If the reviewer has an alternative UFT strategy in mind, we would consider evaluating it for the final version of the paper.
>
> We hope these clarifications and additional HotpotQA results address the reviewer's concerns. Thank you for the feedback!

---

### Review · Reviewer_5fpz · 2025-05-02

**Summary Of Contributions:**

The paper proposes to combine prompt distillation techniques with synthetic data generation in order to inject new knowledge in a language model. Specifically, given a collection of documents from which we would like to extract knowledge, the authors propose to

1. generate Q/A pairs conditioned on a given document
2. predict the answer with a teacher model, which has the document in context
3. predict the answer with a student model that doesn't have access to the document, but has trainable LoRA parameters in which to store document information
4. minimize the KL between (frozen) teacher and student models.

The authors evaluate their approach on the SquadShift datasets (where the questions were rephrased to be more amenable to closed-book evaluation). Results suggest that the proposed approach performs on par with RAG, while being cheaper & faster, and can also yield additional gains when used in tandem with RAG.
Moreover, the authors observe that using a stronger model for data generation or as a teacher in general doesn't lead to better performance. Finally, the authors run several ablations, namely on the impact of the temperature parameters (for both distillation and Q/A generation), and explore using a replay-based regularization objective to limit catastrophic forgetting during knowledge learning.

**Audience:**

Yes

**Broader Impact Concerns:**

No concerns beyond standard concerns resulting from LLM finetuning

**Claims And Evidence:**

Yes

**Requested Changes:**

Below I list a few changes, that I believe would make the paper stronger, but are not critical for acceptance; I believe the current paper as-is meets the bar for TMLR

1. Evaluating the proposed method on longer-context Q/A tasks, or even another Q/A dataset; simply to show that the proposed method is not exploiting quirks specific to SquadShifts

**Strengths And Weaknesses:**

### Strengths
1. The paper proposes a simple, elegant idea and shows that it performs better that standard finetuning approaches in the closed-book setting.
2. The paper is clearly written, and easy to follow. The authors did a good job at contextualizing the work, which touches on many related topics
3. The experimental section is well built; the relevant ablations are provided to isolate the impact of each component. Moreover, the authors evaluate multiple base models and the results are consistent among them.

### Weaknesses
1. The main limitation of this work, is with respect to the dataset used. The documents are relatively short (<2k tokens on average), and it remains unclear if a prompt distillation based approach can scale to longer documents. It would be good if the authors could discuss ways to adapt their method to settings where the document goes beyond the context window of the LLM (e.g. by chunking the document)
2. What is PD XL exactly ? Is it PD with the three changes added (more training questions + Bonito + 1 epoch) ?

---

> ### Author Response · Authors · 2025-05-20
>
> We thank the reviewer for their positive feedback and for recognizing the simplicity and elegance of Prompt Distillation (PD), the paper's clarity, and the thoroughness of our ablations.
>
> 1. W1 & RC1: Document length/evaluation on another dataset
>
> The reviewer raises a valid point regarding the relatively short document lengths in Squadshifts. The training signal in PD (and SFT) comes from the QA pairs, which typically focus on specific facts or pieces of information within a limited span of the text. For long documents, chunking is a practical approach. The distillation process should be effective as long as the model can generate relevant QA pairs and the teacher generates reasonable soft targets. However, the model might still struggle if answering the test questions requires synthesizing information from multiple chunks (different parts of the original long document). This is the challenge that multi-hop reasoning benchmarks evaluate. To demonstrate the effectiveness of PD in this setting and to address the request for evaluation on another dataset, we conducted experiments on HotpotQA. We used a closed-book setup with 1000 questions, corresponding to approximately 10,000 documents (paragraphs). For PD and SFT, training data (QA pairs) was generated from individual documents. In contrast, test questions require combining information across multiple documents. Our preliminary HotpotQA results with Llama-3-8B-Instruct are as follows:
>
> | Method                    | Score (%) |
> |---------------------------|-----------|
> | Llama-3 + Oracle RAG      | 87.1      |
> | Llama-3 + PD + RAG        | 82.1      |
> | Llama-3 + SFT + RAG       | 79.6      |
> | Llama-3 + RAG             | 79.5      |
> | Llama-3 + PD              | 73.7      |
> | Llama-3 + SFT             | 68.9      |
> | Llama-3-8B-Instruct       | 53.0      |
>
> These results show that PD continues to perform strongly, outperforming SFT and enhancing RAG, even when the task requires integrating knowledge that would be found in different chunks of a longer document. Our results suggest that PD is not merely exploiting quirks in Squadshifts. It can generalize to more complex reasoning scenarios. We plan to include these results and results with Qwen in the revised manuscript.
>
> 2. W2: Definition of PD XL.
>
> PD XL refers to PD with three modifications: (1) increasing the number of training questions, (2) using our reproduction of Bonito for question generation, and (3) limiting training to a single epoch. This PD XL configuration was used with the Tülu 3 dataset for regularization. We will make it clear in the paper that these are the changes made. We still used LLama-3-8B-Instruct as the base model.
>
> We believe these clarifications and new experimental results on HotpotQA address the reviewer's points. Thank you for your valuable feedback!

---

> > ### Comment · Reviewer_5fpz · 2025-06-02
> > **Thank you for the updated version**
> >
> > Thank you for the new experiment on HotpotQA. Given that this dataset is open-domain, and according to the authors information must be aggregated from disjoint paragraphs, I am satisfied with the current results.

---

### Review · Reviewer_xs68 · 2025-05-14

**Summary Of Contributions:**

This paper proposes a self-distillation-based technique called Prompt Distillation (PD) for injecting new knowledge into LLMs from unstructured documents, without relying on external retrieval at inference time or larger teacher models for generating synthetic data. Instead of using hard-label supervised fine-tuning with QA pairs, PD uses a single LLM to generate QA pairs with access to context knowledge, and then train the same model (with LoRA) to imitate the full token distribution (logits) from this context-augmented version without accessing c at test time, minimizing the KL divergence between the token-level output distributions of two models. The student and teacher are instantiated from the same base model, with the student equipped with a LoRA adapter, ensuring no additional memory overhead compared to SFT with LoRA. Experimental results using Llama-3 and Qwen 2.5 on the SQuADshifts benchmark show that PD matches or outperforms RAG in knowledge injection tasks.

**Audience:**

Yes

**Broader Impact Concerns:**

I don't think there are any concrete concerns on the ethical implications.

**Claims And Evidence:**

Yes

**Requested Changes:**

1. **Broader evaluation would strengthen the claims**: I believe that adding evaluations on more datasets would better support the paper’s conclusions. For example, [Ovadia et al. (2023)](https://arxiv.org/abs/2312.05934) include MMLU in their evaluation. While MMLU is also relatively simple, it does not come with associated gold context documents, and thus may provide a more realistic testbed for the proposed method, especially in relation to the scalability concerns raised in Weakness 3.

2. **Need for standard QA metrics**: I would also encourage the authors to report results using standard QA metrics such as exact match (EM) or F1, in addition to the LLM-as-a-judge evaluation. While model-based grading can provide useful signal, it introduces noise and complicates comparisons with prior work. Given that SQuADshifts is derived from span-based QA, reporting EM/F1 would offer a more grounded and interpretable evaluation.

**Strengths And Weaknesses:**

### Strengths
- **Technical novelty**: While self-distillation with hard-label supervision and synthetic QA data has been used for knowledge injection in LLMs, the proposed approach, context-guided self-distillation using full token-level knowledge distillation (KL divergence) is, to my knowledge, novel and well-motivated.

- **Strong empirical findings**: The finding that PD significantly outperforms standard supervised fine-tuning (SFT) is particularly interesting, especially given that self-distillation in practice often reduces to SFT with hard output labels. These results may have broader implications for improving parameter-efficient knowledge injection, even in RAG scenarios.

- **Training efficiency**: The use of LoRA adapters is well-motivated, and the design where the same underlying model serves as both teacher and student by toggling the adapter avoids the overhead of running two separate models. This contributes to a memory- and compute-efficient training setup.

### Weaknesses
- **Limited evaluation on a single synthetic QA dataset**: One major limitation is that all evaluations are conducted on a synthetic QA benchmark derived from SQuADshifts. While useful for controlled comparisons, this dataset does not capture the complexity or diversity of real-world knowledge injection scenarios. It remains unclear how well PD would perform in more complex tasks, such as multi-hop reasoning, summarization, or open-ended generation.

- **Evaluation based on LLM-as-a-judge**: The authors rely on LLM-as-a-judge metrics instead of reporting standard string-matching metrics like exact match (EM) or F1, which are commonly used for SQuAD-like benchmarks including the original SQuADShift evaluation. While the LLM-as-a-judge setup is reasonable, it introduces additional uncertainty, biases and limits comparability to prior work. With careful prompting, it is possible to elicit short-form answers for evaluation, as done in works like REPLUG (Shi et al., 2024), and I thought authors should also report such standard string-matching based evaluations, especially given that currently evaluations are only done in a short-form QA dataset.

- **Dependence on `c` and generalizability / scalability concerns**: To generate training examples, the method requires access to a knowledge context `c` (e.g., a document) that is associated with a query `q`. In realistic settings, however, such gold `c-q` alignments are not available. While it is possible to generate synthetic QA pairs for each document in the large corpus `C`, this may be computationally expensive or infeasible for large-scale corpora (e.g., tens of millions of documents), and may not reflect the true query distribution. The effectiveness of PD in such low-resource or noisy settings remains unclear.

---

> ### Author Response · Authors · 2025-05-20
>
> We thank the reviewer for their insightful feedback and for recognizing the technical novelty, strong empirical findings, and training efficiency of Prompt Distillation (PD). We appreciate the constructive suggestions for improvement and have taken steps to address the requested changes.
>
> 1. W1 & RC1: Limited evaluation on a single synthetic QA dataset / Broader evaluation
>
> We agree that evaluation on more complex datasets would strengthen the claims. To address this, we have conducted preliminary experiments on HotpotQA, a benchmark for multi-hop reasoning. We used a closed-book setup with 1000 questions, corresponding to approximately 10,000 documents (paragraphs). For PD and SFT, training data (QA pairs) was generated from individual documents. In contrast, test questions require synthesizing information across multiple documents. Hence, this evaluates generalization to more complex, out-of-distribution reasoning tasks. Our preliminary HotpotQA results with Llama-3-8B-Instruct are as follows:
>
> | Method | Score (%) |
> |---------------------------|-----------|
> | Llama-3 + Oracle RAG | 87.1 |
> | Llama-3 + PD + RAG | 82.1 |
> | Llama-3 + SFT + RAG | 79.6 |
> | Llama-3 + RAG | 79.5 |
> | Llama-3 + PD | 73.7 |
> | Llama-3 + SFT | 68.9 |
> | Llama-3-8B-Instruct | 53.0 |
>
> These results demonstrate that PD outperforms SFT in closed-book settings and when combined with RAG, even on this challenging multi-hop task. PD with RAG outperforms standard RAG, suggesting that the PD-trained model can compensate for imperfect retrievals through internalized knowledge. We plan to include these results and results with Qwen in the revised manuscript.
>
> 2. W2 & RC2. Evaluation based on LLM-as-a-judge
>
> We acknowledge the value of standard metrics like F1 for comparability. We chose LLM-as-a-judge to assess the correctness of the information in a closed-book setting, where the generated answers can be phrased differently from gold-standard short answers, which could make results with string-matching metrics slightly misleading. We used two different LLM families for grading to mitigate grader bias and found consistent results. However, we have computed F1 scores for our Llama experiments on Squadshifts to address the reviewer's request. The F1 scores below (as percentages) were calculated on the long-form answers generated by the models. The verbosity of the answers naturally leads to high recall but very low precision, which affects the F1 scores. Hence, these results should be interpreted as supporting evidence for the reliability of our LLM-as-a-judge-based grading.
>
> F1 Scores (%):
> | Method | Amazon | New Wiki | NYT | Reddit |
> |--------------------------------|----------|----------|----------|----------|
> | Prompt Distillation | **19.8** | **24.7** | **23.3** | **15.4** |
> | Supervised Fine-Tuning | 17.7 | 23.2 | 21.8 | 13.3 |
> | Unsupervised Fine-Tuning | 3.9 | 6.1 | 5.8 | 3.7 |
> | Llama-3-8B-Instruct | 2.3 | 4.5 | 3.6 | 2.4 |
> | Prompt Distillation + RAG | **21.9** | **26.7** | **24.8** | 16.5 |
> | SFT w/ Distractors + RAG | 19.6 | 23.6 | 23.0 | 15.8 |
> | Llama-3-8B-Instruct + RAG | 18.1 | 24.8 | 22.6 | **17.2** |
>
> As shown, these scores mirror those observed with LLM-as-a-judge. We will evaluate prompting strategies to elicit concise, short-form answers for standard F1 evaluation and plan to include these comparisons in the final version.
>
> 3. W3. Dependence on c, generalizability, and scalability.
>
> This is a very valid point, and RAG is most likely essential for massive datasets. However, as our results indicate, supplementing RAG with PD can improve performance. It's worth noting that standard SFT baselines also require such QA pairs for knowledge injection. As our results in Figure 3 indicate, PD demonstrates superior data efficiency compared to SFT. It achieves better performance with the same amount of training data (or comparable performance with less). This efficiency can be crucial when there are budget constraints and the overhead of QA pair generation must be minimized. We acknowledge that scaling to corpora with tens of millions of documents is beyond the scope of this work, and our approach targets scenarios where generating sufficient QA data is feasible. We did not explicitly test PD in noisy settings (such as conflicting information in the documents). However, PD's soft targets from the teacher (compared to hard labels in SFT) should offer resilience to noise in the generated answers. The student learns from the teacher's entire distribution, which can implicitly down-weight noisy tokens if the teacher assigns them a lower probability.
>
> We believe these new results and clarifications address the reviewer's primary concerns. We will include the new HotpotQA results and F1 scores in the final version of the manuscript. Thank you for your valuable feedback!

---

> > ### Comment · Reviewer_xs68 · 2025-05-22
> > **Thank you for updates and new results**
> >
> > Thank you so much for sharing the updated results. I really appreciate the effort you've put into the evaluation. That said, I was quite surprised to see such low F1 scores on SQuAD Shift. While I understand that your evaluation setup differs from the original SQuAD Shift - framing it more as open-domain QA rather than reading comprehension with provided passages - a 20% F1 score still seems quite low.
> >
> > For context, even relatively simple retrieval-augmented baselines such as the RAG model from [Ram et al. (2023)](https://arxiv.org/pdf/2302.00083), using older models like LLaMA-1, achieve around 30% EM on Natural Questions and 60% on TriviaQA. These benchmarks use Exact Match, which is an even stricter metric than F1. Given the similarity in task format, it's surprising that both the baselines and proposed methods here are hovering around 20% F1 - despite reporting 80–90% accuracy using the "LLM-as-a-judge" evaluation.
> >
> > If the low F1 scores are primarily due to verbosity or formatting differences, could you additionally report an answer match metric (e.g., [Mallen et al., 2023](https://arxiv.org/abs/2212.10511)), which checks whether any of the gold answers appear in the generated response? That could help clarify whether the models are producing correct content but in a mismatched format.
> >
> > More broadly, this discrepancy between the standard F1 scores and the high LLM-as-a-judge ratings raises concerns about the reliability of the latter, especially since it’s not the standard evaluation approach for this dataset. I'd be interested in hearing your thoughts on this gap.

---

> > > ### Author Response · Authors · 2025-05-23
> > >
> > > Thank you for your continued engagement and excellent suggestion to report an answer match metric. We agree that the F1 scores reported in our previous response are very low compared to the LLM-as-a-judge metrics, and we understand your concern about this discrepancy. As we discussed, this is primarily due to the verbosity of answers from instruction-tuned LLMs, which heavily penalizes precision in F1 calculations despite the models often including the correct content (as indicated by high word overlap recall). The gold answers in SQuADShifts are typically very concise, leading to this mismatch.
> > >
> > > Following your proposal and the approach of Mallen et al. (2023), we have now computed answer match scores by checking if any gold answers appear as a substring in the generated response after simple normalization. These new scores help bridge the gap you identified. Below are the answer match scores for our main experiments:
> > >
> > > Base Model: Llama‑3‑8B‑Instruct
> > > | Method                    |  Amazon  | New Wiki |    NYT   |  Reddit  |
> > > | ------------------------- | :------: | :------: | :------: | :------: |
> > > | Prompt Distillation       | **59.1** | **75.6** | **79.6** | **62.1** |
> > > | Supervised Fine‑Tuning    |   51.8   |   71.0   |   74.5   |   53.4   |
> > > | Unsupervised Fine‑Tuning  |   27.2   |   51.1   |   46.9   |   27.7   |
> > > | Llama‑3‑8B‑Instruct       |   12.5   |   31.8   |   23.0   |   12.3   |
> > > | Prompt Distillation + RAG | **69.9** | **86.1** | **87.7** | **68.6** |
> > > | SFT w/ Distractors + RAG  |   65.3   |   78.1   |   84.0   |   66.1   |
> > > | Llama‑3‑8B‑Instruct + RAG |   65.5   |   83.8   |   85.7   |   58.7   |
> > >
> > > Base Model: Qwen2.5‑14B‑Instruct
> > > | Method                     |  Amazon  | New Wiki |    NYT   |  Reddit  |
> > > | -------------------------- | :------: | :------: | :------: | :------: |
> > > | Prompt Distillation        | **60.0** | **72.8** | **79.0** | **62.8** |
> > > | Supervised Fine‑Tuning     |   53.3   |   68.0   |   74.1   |   57.1   |
> > > | Qwen2.5‑14B‑Instruct       |   11.4   |   36.5   |   21.7   |   13.9   |
> > > | Prompt Distillation + RAG  | **67.0** | **80.3** |   84.5   |   67.4   |
> > > | SFT + RAG                  |   66.9   |   79.9   | **84.7** | **68.0** |
> > > | Qwen2.5‑14B‑Instruct + RAG |   65.5   |   77.2   |   82.9   |   66.2   |
> > >
> > > Base Model: Qwen2.5‑3B‑Instruct
> > > | Method                    |  Amazon  | New Wiki |    NYT   |  Reddit  |
> > > | ------------------------- | :------: | :------: | :------: | :------: |
> > > | Prompt Distillation       | **52.7** | **68.1** | **72.8** | **52.7** |
> > > | Supervised Fine‑Tuning    |   48.5   |   65.8   |   66.1   |   50.2   |
> > > | Qwen2.5‑3B‑Instruct       |   10.3   |   28.9   |   14.4   |   11.3   |
> > > | Prompt Distillation + RAG | **67.5** | **79.9** | **84.4** | **66.9** |
> > > | SFT + RAG                 |   62.1   |   77.3   |   80.4   |   63.4   |
> > > | Qwen2.5‑3B‑Instruct + RAG |   66.3   |   77.4   |   80.7   |   65.7   |
> > >
> > > Llama‑3‑8B‑Instruct + Tülu 3 Dataset
> > > | Method                          |  Amazon  | New Wiki |    NYT   |  Reddit  |
> > > | ------------------------------- | :------: | :------: | :------: | :------: |
> > > | Prompt Distillation             |   59.1   |   75.6   |   79.6   | **64.0** |
> > > | Prompt Distillation + Tülu 3    |   59.2   |   76.3   |   79.6   |   61.1   |
> > > | Prompt Distillation XL + Tülu 3 |   63.5   |   79.2   |   83.7   |   62.9   |
> > > | Llama‑3‑8B‑Instruct + RAG       | **65.5** | **83.8** | **85.7** |   58.7   |
> > >
> > > These substring match scores are significantly higher than the F1 scores. The models frequently embed the correct answer string within their outputs, and the low F1 scores were an artifact of the precision penalty from verbosity. The gap to our LLM-as-a-judge evaluations is much smaller. For instance, with Llama-3-8B-Instruct, the average LLM-graded correctness for PD was 88.4 %, while the average substring match was 69.1 %. For PD+RAG, the LLM-graded correctness was 90.0%, with an average substring match of 78.1 %. Hence, the mean grading gap is around 15 %, which we believe is natural and explainable. LLM-as-a-judge can be more lenient, recognizing semantic equivalence instead of focusing on the choice of words. Substring match is still an inherently strict lexical metric (e.g., is the gold answer 6 or "six"). We hope this alleviates concerns about the reliability of the LLM-as-a-judge for assessing correctness.
> > >
> > > Furthermore, we maintain that RAG-based methods enjoy an in-built advantage in this evaluation setting as they can directly copy answers from the provided context. Closed-book PD XL approaching the performance of the RAG baseline despite the natural disadvantage is encouraging. Most importantly, the relative ranking and observed improvements of PD over baselines remain consistent across both substring match and LLM-as-a-judge evaluations. Thank you for your feedback, which helps us strengthen the evaluation of our method.

---

### Author Response · Authors · 2025-05-27

We thank all the reviewers for their insightful feedback, which has helped us improve our paper. We are encouraged that the reviewers recognize the technical novelty, elegance, practicality, strong empirical evaluation and findings, and training efficiency of Prompt Distillation, as well as the overall clarity of our work.

We have uploaded a revised manuscript incorporating the following changes in response to the reviewers' suggestions:

1. Added HotpotQA Evaluations (Reviewers xs68, 5fpz, 67aR): We have incorporated a new section, Section 4.7, presenting evaluations on the HotpotQA dataset, a challenging multi-hop reasoning benchmark. These experiments, conducted with all three base models, extend our evaluation beyond Squadshifts to a new domain and assess PD's ability to improve reasoning with knowledge integrated from multiple documents. As shown in Table 7, PD demonstrates substantial improvements in the closed-book setting, outperforming SFT. Furthermore, when combined with RAG, PD significantly surpasses the base model's performance with RAG alone. Minor modifications have also been made to the introduction and conclusions to reflect these findings.

2. Incorporated Substring Match Grading (Reviewer xs68): To complement our LLM-as-a-judge evaluation, we have implemented substring match-based grading, following Mallen et al. (2023). This method, discussed in the new Appendix N, marks an answer as correct if any of the gold answers appear as a substring in the generated response after simple normalization. This alternative metric confirmed the same relative performance trends observed with our primary LLM-based evaluation. The main text in Section 4.2 (Answer Grading) has been updated to reference this additional evaluation.

3. Clarified PD XL Configuration (Reviewer 5fpz): In response to this request, we have refined the discussion of the Prompt Distillation XL (PD XL) configuration in Section 4.6. The text now more clearly describes the similarities and differences between the methods.

## References

Mallen, A., Asai, A., Zhong, V., Das, R., Khashabi, D., & Hajishirzi, H. (2023). When not to trust language models: Investigating effectiveness of parametric and non-parametric memories.

---

### Decision · Action_Editor_JFKM · 2025-07-11

**Recommendation:** Accept as is

**Audience:**

Yes

**Audience Explanation:**

Definitely audiences interested in parameter-efficient fine-tuning, prompt compression, synthetic data generation, long-context generation but also knowledge representation. The paper will generate definitely interesting future work.

**Claims And Evidence:**

Yes

**Claims Explanation:**

Prompt Distillation (PD) beats standard supervised fine-tuning (SFT) and rivals retrieval-augmented generation (RAG) for knowledge injection, PD + RAG is shown to work better than RAG and scaling up facts to be integrated in the PD training procedure outperform RAG. This has been shown both with LLM as a judge and automated metrics (as suggested by a reviewer). The method is benchmarked on multiple dataset and with multiple backbone models, the evidence is convincing and the method is scientifically appealing. Might definitely generate interesting future work.